The EMBO Journal (2013) 32, 2848–2860
www.embojournal.org

THE
EMBO
JOURNAL

# TRIAD1 and HHARI bind to and are activated by distinct neddylated Cullin-RING ligase complexes

**Ian R Kelsall[1,2], David M Duda[3], Jennifer L Olszewski[3], Kay Hofmann[4], Axel Knebel[1,2], Frédéric Langevin[5], Nicola Wood[1,2], Melanie Wightman[1,2], Brenda A Schulman[3] and Arno F Alpi[1,2,*]**

[1]Scottish Institute for Cell Signalling, College of Life Sciences, University of Dundee, Dundee, UK, [2]Medical Research Council Protein Phosphorylation and Ubiquitylation Unit, College of Life Sciences, University of Dundee, Dundee, UK, [3]Department of Structural Biology, Howard Hughes Medical Institute, St. Jude Children's Research Hospital, Memphis, TN, USA, [4]Institute of Genetics, University of Cologne, Cologne, Germany and [5]Division of Protein and Nucleic Acid Chemistry, Medical Research Council Laboratory of Molecular Biology, Cambridge, UK

RING (Really Interesting New Gene)-in-between-RING (RBR) enzymes are a distinct class of E3 ubiquitin ligases possessing a cluster of three zinc-binding domains that cooperate to catalyse ubiquitin transfer. The regulation and biological function for most members of the RBR ligases is not known, and all RBR E3s characterized to date are auto-inhibited for in vitro ubiquitylation. Here, we show that TRIAD1 and HHARI, two members of the Ariadne subfamily ligases, associate with distinct neddylated Cullin-RING ligase (CRL) complexes. In comparison to the modest E3 ligase activity displayed by isolated TRIAD1 or HHARI, binding of the cognate neddylated CRL to TRIAD1 or HHARI greatly stimulates RBR ligase activity in vitro, as determined by auto-ubiquitylation, their ability to stimulate dissociation of a thioester-linked UBCH7~ubiquitin intermediate, and reactivity with ubiquitin-vinyl methyl ester. Moreover, genetic evidence shows that RBR ligase activity impacts both the levels and activities of neddylated CRLs in vivo. Cumulatively, our work proposes a conserved mechanism of CRL-induced Ariadne RBR ligase activation and further suggests a reciprocal role of this special class of RBRs as regulators of distinct CRLs.

The EMBO Journal (2013) **32,** 2848–2860. doi:10.1038/emboj.2013.209; Published online 27 September 2013
Subject Categories: proteins
Keywords: auto-inhibition; Cullin-RING ligases; HHARI; RBR E3 ubiquitin ligases; TRIAD1

## Introduction

The RING (Really Interesting New Gene) family comprises the largest group of E3 ubiquitin ligases with >600 members

*Corresponding author. Medical Research Council Protein Phosphorylation and Ubiquitylation Unit, College of Life Sciences, University of Dundee, Dow Street, Dundee DD1 5EH, UK. Tel.: +44 1382384999; Fax: +44 1382223778; E-mail: a.f.alpi@dundee.ac.uk

in mammals (Deshaies and Joazeiro, 2009). Among these, the multisubunit Cullin-RING ligases (CRLs) control ubiquitin-mediated degradation of proteins in numerous biological processes (Petroski and Deshaies, 2005). CRLs have a modular composition and are assembled by one of the eight elongated cullin protein scaffolds (CUL1, 2, 3, 4A, 4B, 5, 7, and 9) each of which binds a RING E3 ligase (RBX1 or RBX2) at the C-terminus and various substrate adapter complexes at the N-terminus. CRL activity is tightly regulated by the reversible ligation of the ubiquitin-like protein NEDD8 to a conserved lysine in the C-terminus of the cullin (Deshaies et al, 2010). Like ubiquitin NEDD8 is initially activated by its E1 activation enzyme APPBP1/UBA3 and transferred to the catalytic cysteine of the E2 (UBE2M or UBE2F in the case of CUL5) (Lammer et al, 1998; Liakopoulos et al, 1998; Osaka et al, 1998; Duda et al, 2008; Huang et al, 2009). The subsequent NEDD8 ligation to the cullin is promoted by the synergistic activity of DCN1 and the RING subunit of the CRL complex (Scott et al, 2010). NEDD8 attachment stimulates the assembly of an intact CRL complex and switches on the ubiquitin ligase activity by repositioning the RING E3 ligase, thus orientating the E2 active site adjacent to the substrate (Duda et al, 2008, 2011; Saha and Deshaies, 2008; Yamoah et al, 2008). NEDD8 is removed from cullins by the COP9 signalosome (CSN) complex, thereby reverting the CRL to an inactive state (Lyapina et al, 2001). Continuous neddylation and de-neddylation therefore plays a crucial part in the dynamic regulation of CRLs, but an intriguing question is whether NEDD8 has other functions besides influencing the assembly and activation state of a CRL. Notably, recent studies have found that NEDD8 dictates binding of a subset of CRLs to UBXN7/p97 complexes to regulate turnover of their ubiquitylation substrates. This raises the possibility that neddylation may direct CRLs to other complexes for additional levels of regulation by the ubiquitin-proteasome system (Bandau et al, 2012; den Besten et al, 2012).

In contrast to CRLs, the class of RING ligases defined by the RING-in-between-RING (RBR) domain is less well studied (Wenzel and Klevit, 2012). Best-known members of this group are PARKIN and HOIP, and defects in their ligase activity are associated with neuronal degenerative diseases and innate immune deficiency, respectively. RBRs are characterized by a cluster of three zinc-binding domains (Eisenhaber et al, 2007; Marin, 2009): (1) the N-terminal RING domain (RING1) coordinates two zinc ions into the canonical double cross-brace RING structure and interacts with the E2; (2) the central IBR (in-between RING) domain also binds to zinc ions, and (3) the C-terminal RING domain (RING2) was revealed by crystallographic studies to display topological distinctions from classical RINGs and in fact assemble a structure similar to IBRs (Duda et al, 2013; Riley et al, 2013; Spratt et al, 2013; Trempe et al, 2013; Wauer and Komander, 2013). A landmark finding was that a conserved cysteine in RING2 forms a transient thioester bond with

ubiquitin, facilitating subsequent ubiquitin ligation to a lysine (Wenzel et al, 2011). However, recent studies have revealed that this RING2 catalytic cysteine is typically masked in full-length forms of RBR E3s (Chaugule et al, 2011; Smit et al, 2012; Stieglitz et al, 2012; Duda et al, 2013; Riley et al, 2013; Spratt et al, 2013; Trempe et al, 2013; Wauer and Komander, 2013), rendering ligase activity auto-inhibited.

In animals, RBR ligases can be subclassified into 12 sub-families based on the phylogenetic relationships of the RBR domains and the conservation of their sequence architecture (Marin, 2009). Ariadne forms the largest, most diverse and oldest of these RBR subfamilies. Ariadne RBRs have in common a C-terminal 'Ariadne' domain in addition to the RBR domain (Marin and Ferrus, 2002). The Ariadne RBRs ari-1 and ari-2 were originally identified in D. melanogaster where they were shown to be important for fly development (Aguilera et al, 2000). The human homologue of ari-1, HHARI, is highly expressed in nuclei, where it is co-localized with nuclear bodies including Cajal, PML, and Lewy bodies, suggesting a nuclear function of HHARI (Parelkar et al, 2012; Elmehdawi et al, 2013). The mammalian homologue of ari-2, TRIAD1, has been implicated in haematopoiesis, specifically in myelopoiesis (Marteijn et al, 2005). Moreover, TRIAD1 is essential for embryogenesis, and TRIAD1-deficient mice die due to a severe and lethal multiorgan immune response (Lin et al, 2012). Despite some evidence for the biological importance, the mechanisms regulating Ariadne RBR ligase function remain poorly understood.

Here, we uncover that two members of the Ariadne sub-family of RBR ligases, TRIAD1 and HHARI, associate with distinct but neddylated CRL complexes. NEDD8-CRL binding greatly stimulated the ubiquitin ligase activities of these Ariadne family RBR E3s in vitro. Moreover, mutations in the RING2 and Ariadne domains of TRIAD1 and HHARI influence the neddylation state of their cognate cullins in vivo. This work suggests a mechanism regulating the RBR ligase activity of TRIAD1 and HHARI, demonstrates a novel function of cullin neddylation, and demonstrates a functional linkage between RBR and CRL E3s.

## Results

### TRIAD1 is an RBR E3 ubiquitin ligase

To better understand the ubiquitin ligase function of the RBR protein TRIAD1, we initially aimed to identify and character-ize the cognate E2 conjugating enzymes. We screened a panel of 34 E2 enzymes to determine which E2 catalysed TRIAD1 auto-ubiquitylation. This screen revealed that UBCH7 was the only conjugating enzyme that significantly catalysed TRIAD1 auto-ubiquitylation (Supplementary Figure S1A). We next employed co-immunoprecipitation experiments to test whether TRIAD1 interacts with UBCH7 in cells. TRIAD1-specific antibodies immunoprecipitated endogeneous com-plexes of TRIAD1 with UBCH7, showing an in vivo associa-tion (Figure 1A) (Markson et al, 2009). In addition, in vitro binding assays with purified recombinant His$_6$-TRIAD1 confirmed direct binding to UBCH7 (Figure 1B). We next investigated which of the three zinc-binding domains in TRIAD1 are required for the interaction with E2 by mutating structurally conserved histidine and cysteine residues pre-dicted to coordinate zinc ions. We observed that mutating a

histidine residue in the RING1 domain to alanine (H158A) abolished binding to UBCH7, whereas mutations in the IBR (C257A) or RING2 (C300A) domain had no such effect (Figure 1C). Cumulatively, these data suggest that UBCH7 is the functional E2 partner of TRIAD1 and further corroborate the current notion that UBCH7 is a physiological E2 for RBR-type E3 ligases.

To further analyse the TRIAD1 E3 ligase activity, we set up in vitro activity assays aimed at monitoring intrinsic function, because to date assays have not been established for any physiologically relevant substrate of TRIAD1. First, we employed a recently described method to assay RBR ligase activity by monitoring RBR E3-dependent dissociation of a UBCH7~ubiquitin thioester intermediate (~ indicates thioester bond) in the presence of free lysine (Wenzel et al, 2011). This assay (hereafter referred to as UBCH7~ubiquitin discharge assay) relies on the distinct inability of UBCH7 to transfer ubiquitin directly to a lysine. Indeed, as observed previously, we found the UBCH7~ubiquitin intermediate to be stable in the presence of lysine. However, addition of TRIAD1 led to disappearance of the UBCH7~ubiquitin thioester intermediate in a concentration- (0–3.6 μM TRIAD1) and time- (0–120 min) dependent manner (Figure 1D and E). Consistent with the notion put forth for other RBR E3s (Wenzel et al, 2011), the data suggest that ubiquitin is transferred from UBCH7 to TRIAD1, and then from TRIAD1 to the lysine in solution. As a second assay, we also analysed TRIAD1 auto-ubiquitylation activity by immunoblotting (Supplementary Figure S1B), and found that this correlated well with UBCH7~ubiquitin discharge data.

Recent studies of RBR ligases have suggested that a conserved RING2 cysteine is required for ubiquitin ligase activity through formation of a transient thioester-linked RBR~ubiquitin intermediate, like that formed by HECT E3s (Wenzel et al, 2011; Smit et al, 2012; Stieglitz et al, 2012; Riley et al, 2013; Spratt et al, 2013; Trempe et al, 2013). Indeed, mutating this conserved Cys310 in TRIAD1 to alanine or serine abolished TRIAD1 auto-ubiquitylation (Figure 1F). Consistent with the higher degree of reactivity observed for other RBR E3s (Wenzel et al, 2011; Smit et al, 2012; Stieglitz et al, 2012; Duda et al, 2013), we were not able to detect an ubiquitin thioester on C310, or the more stable ubiquitin oxyester, which is predicted to form with the C310S TRIAD1 mutant. Additional analysis of TRIAD1 auto-ubiquitylation showed that UBCH7 reactions preferentially produced K6- and K48-linked ubiquitin chains on TRIAD1 (Figure 1G).

### TRIAD1 specifically associates with cullin-5

To gain insights into TRIAD1 functions, we performed mass spectrometry analysis of anti-GFP immunoprecipitates from lysates of cells stably expressing GFP or GFP-tagged TRIAD1. The comparative analysis of these immunoprecipitates re-vealed cullin-5 (CUL5) and UBCH7 as specific TRIAD1 inter-action partners (Figure 2A). The formation of endogeneous TRIAD1/CUL5 complex was confirmed by immunoblotting (Figure 2B). Only a small fraction of TRIAD1 and CUL5 forms a complex, and we further noted that only the slower migrating, neddylated form of CUL5 (CUL5-N8) co-precipi-tated with TRIAD1 (see below). CUL5 is one of eight cullin scaffold proteins, so we tested whether other cullins can also associate with TRIAD1. Cell lines expressing N-terminally HA-tagged CUL1, 2, 3, 4A, 4B, 5, or 7 were generated and

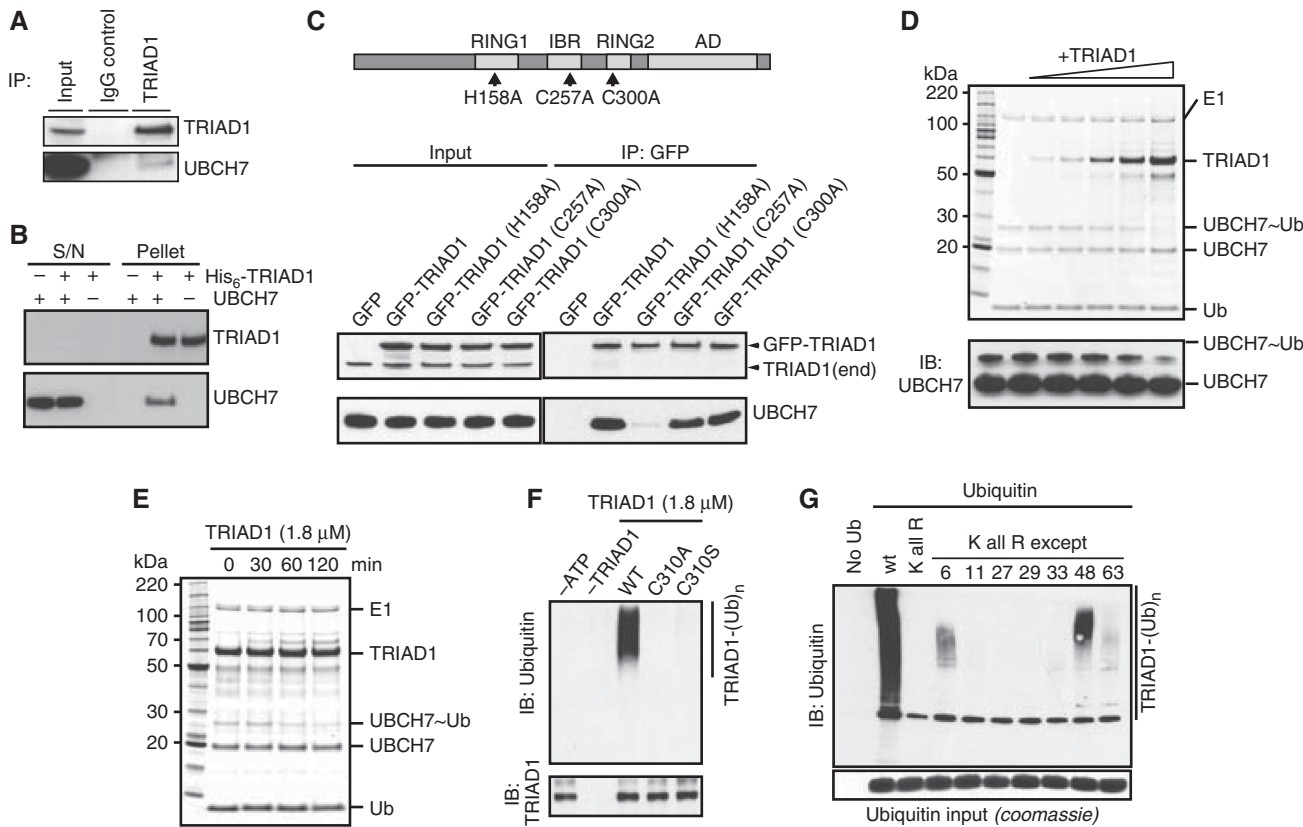

**Figure 1** Characterization of TRIAD1 ubiquitin ligase activity. (**A**) HEK293 lysates were subjected to immunoprecipitation with pre-immune IgG or anti-TRIAD1 antibody and immunoblotted with the indicated antibodies. (**B**) *In vitro* nickel-NTA precipitations were performed using recombinant His$_6$-TRIAD1 and untagged UBCH7. Binding was detected by immunoblot analysis of the assay supernatants (S/N) and pellets with the indicated antibodies. (**C**) GFP-tagged wild-type or mutant TRIAD1 containing the amino-acid substitutions indicated in the upper schematic (showing the domain structure of the protein, including the Ariadne domain) was stably expressed in HEK293 cells and immunoadsorbed using anti-GFP agarose. The inputs and immunopellets were immunoblotted as indicated. The anti-TRIAD1 antibody detects both the endogeneous and slower migrating exogeneous proteins. (**D**) UBCH7 ∼ ubiquitin thioester was incubated with increasing concentrations of TRIAD1 (0.15–3.6 μM) at 37°C for 60 min. Reaction products were resolved on non-reducing SDS–PAGE gels and visualized by SimplyBlue staining (upper panel) or immunoblot analysis using anti-UBCH7 antibody. (**E**) UBCH7 ∼ ubiquitin hydrolysis was assayed in the presence of 1.8 μM TRIAD1 at 37°C for indicated time points and visualized by SimplyBlue staining. (**F**) TRIAD1 auto-ubiquitylation assay ($t = 90$ min) with UBCH7 and the indicated mutants in the TRIAD1 RING2 domain. Also included are control assays lacking ATP or TRIAD1. (**G**) TRIAD1 auto-ubiquitylation ($t = 90$ min) using UBCH7 with WT and mutant forms of ubiquitin to establish ubiquitin linkage preference. A lysine-less ubiquitin (K all R) was used, as were single-lysine versions in which six of ubiquitin's seven lysines were mutated to arginine (K all R except) with the number representing the remaining Lys residue. Upper panel represents an anti-ubiquitin immunoblot; lower panel is SimplyBlue stained to determine equal ubiquitin input.

used for anti-HA immunoprecipitations. Only HA-CUL5 pre-cipitated TRIAD1 indicating a selective interaction of TRIAD1 with CUL5 (Figure 2C).

To further characterize the TRIAD1/CUL5 interaction, we aimed to identify regions of TRIAD1 required to bind CUL5 in cells. A set of GFP-TRIAD1 truncations was analysed for the ability to co-precipitate CUL5 (Supplementary Figure S2A). Deletion of the N-terminal 60 or 120 amino acids completely eliminated the binding of TRIAD1 to CUL5 but did not affect interaction with UBCH7 (Figure 2D). This region contains a conserved stretch of negatively charged amino acids (aa 1–60) (Figure 2E), essential for TRIAD1's interaction with CUL5 in cells (Figure 2D). Hence, to identify a potential corresponding binding site on CUL5 we focussed our analysis on the recently described 'basic canyon' in the C-terminal part of CUL5 (Kleiger *et al*, 2009). This positively charged groove on the convex side of the cullin has been reported to bind to the acidic tail of the E2 enzyme Cdc34, although patterns of conservation suggest that multiple CRL cofactors may engage this particular feature. Replacing conserved basic

residues in HA-CUL5 with acidic residues (glutamates) abolished the interaction with TRIAD1 (Figure 2F). We further noted that more of the neddylated form of CUL5 as well as more of the Elongin-B and Elongin-C substrate adaptor proteins, the essential subunits required for the binding of substrates to CUL5-based CRL complexes, co-precipitated with these mutant CUL5 versions.

### TRIAD1/CUL5 interaction depends on NEDD8

The co-IP experiment presented in Figure 2B indicated that TRIAD1 interacts specifically with the slower migrating, neddylated form of CUL5. To test whether CUL5 neddylation is required for this interaction, we compared the TRIAD1/CUL5 association in the presence or absence of the NEDD8 E1-enzyme inhibitor MLN4924 (Soucy *et al*, 2009). Cells were either mock treated or treated overnight with MLN4924, efficiently depleting the pool of neddylated CUL5 (CUL5-N8) (Figure 3A). In the MLN4924-treated cells, TRIAD1's interaction with CUL5, and its adapters Elongin-B/C, was completely abolished, whereas GFP-TRIAD1 precipitated

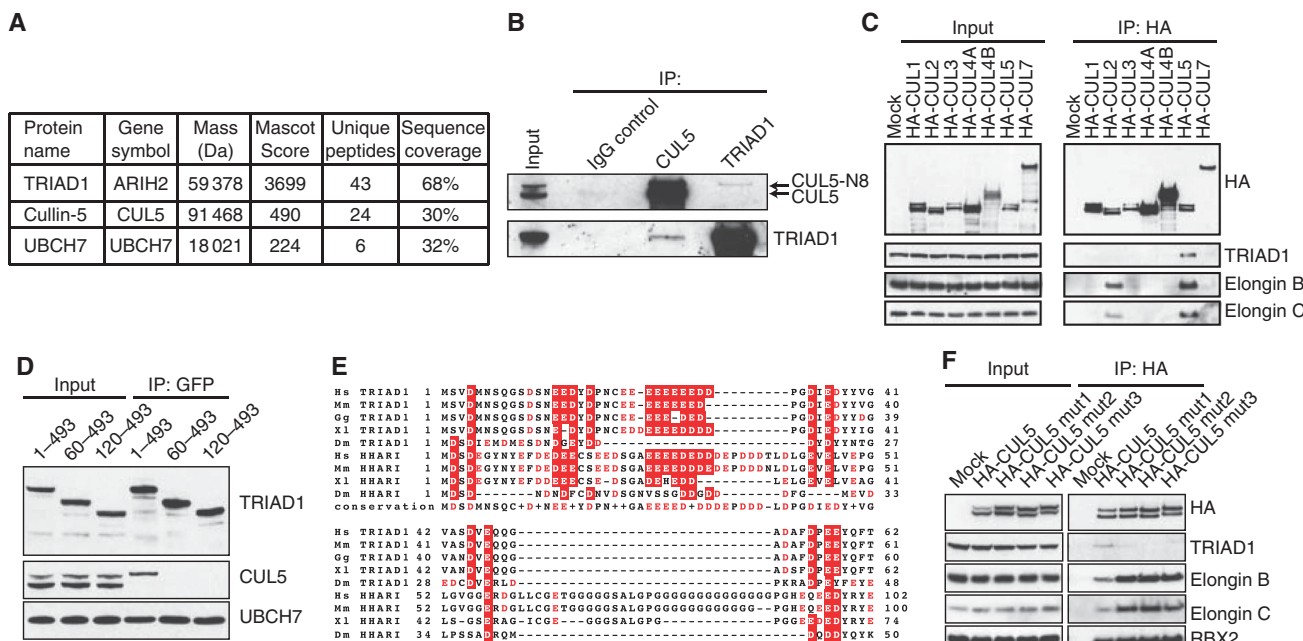

**Figure 2** TRIAD1 associates with cullin-5. (**A**) Table showing proteins identified by mass spectrometry in GFP immunoprecipitates from a GFP-TRIAD1 expressing cell line. (**B**) Immunoadsorption of endogeneous CUL5 and TRIAD1 from HEK293 cell lysates, followed by immunoblotting with the indicated antibodies. The slower migrating of the two CUL5-reactive bands is the neddylated form of the protein (CUL5-N8). (**C**) Immunoblots of anti-HA immunopellets and cell lysates from HEK293 cells stably expressing the indicated HA-tagged cullins. (**D**) Anti-GFP immunopellets and cell lysates from HEK293 cells stably expressing the indicated GFP-tagged TRIAD1 truncation constructs. (**E**) Alignment of the N-terminus of human (Hs) TRIAD1 with its mouse (Mm), chicken (Gg), *Xenopus* (Xl) and *Drosophila* (Dm) homologues, as well as the closely related ligase HHARI from the same species. (**F**) Wild-type or mutant HA-CUL5 was immunoprecipitated from stably transfected HEK293 cells and immunoblotted with the indicated antibodies. HA-CUL5 mut1: R417E, K418E, K423E, K424E mutant; HA-CUL5 mut2: R417E, K423E, R683E mutant; HA-CUL5 mut3: K676E, K679E, K682E, R683E mutant.

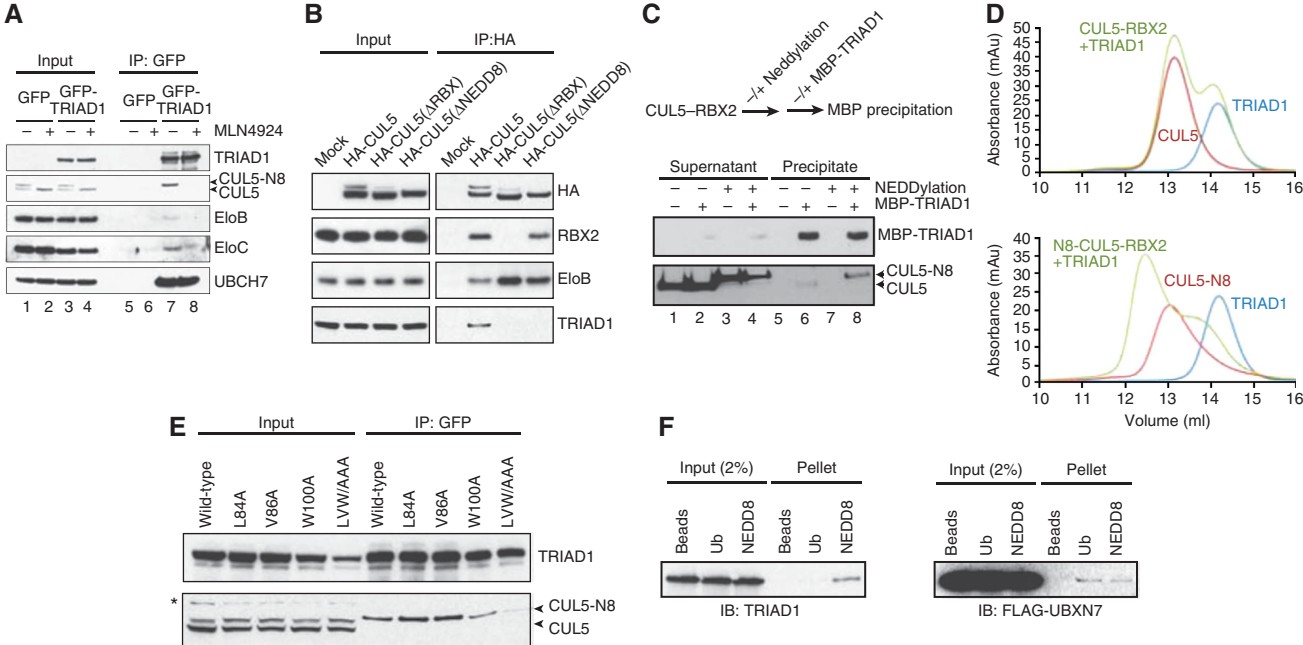

**Figure 3** The TRIAD1/CUL5 interaction is neddylation dependent. (**A**) HEK293 cells stably expressing GFP-TRIAD1 or GFP-only control were treated overnight with 1 µM MLN4924 before immunoprecipitation with anti-GFP antibodies and immunoblotting as indicated. (**B**) Immunoblots of cell lysates and anti-HA immunopellets from HEK293 cells stably expressing HA-tagged WT and mutant CUL5. (**C**) Recombinant MBP-TRIAD1 was used to capture non-neddylated and neddylated recombinant CUL5–RBX2 complex in an *in vitro* binding assay. (**D**) Gel filtration chromatography analysis of TRIAD1 mixed either with non-neddylated (upper graph) or with neddylated CUL5–RBX2 (lower graph). (**E**) GFP-tagged WT and mutant TRIAD1 were immunoprecipitated from HEK293 cells and immunoblotted to detect CUL5 binding. LVW/AAA refers to the triple mutant L84A/V86A/W100A. *Indicates a non-specific band. (**F**) NEDD8- or ubiquitin-agarose beads (or beads only control) were incubated with recombinant wild-type TRIAD1 (left panel) or FLAG-UBXN7 (right panel). Binding was detected by immunoblotting.

CUL5 in mock-treated cells (compare lanes 7 and 8) (Figure 3A). TRIAD1 binding to UBCH7 was not affected upon treatment with MLN4924.

MLN4924 potentially inhibits all neddylation and the observed dependency of the TRIAD1/CUL5-N8 interaction on neddylation might be indirect. To test for a direct contribution of NEDD8 in the TRIAD1/CUL5 interaction, we stably expressed, in HEK293 cells, mutant versions of HA-tagged CUL5 that have previously been shown to abolish CUL5 neddylation by different means. These mutants either had an arginine replacement for the Lys724 site of neddylation (ΔNEDD8) or a short deletion (aa 566–582) that abolished the RBX2 interaction required for neddylation (ΔRBX) (Yu *et al*, 2003). Input samples confirmed that neddylation was defective for both the HA-CUL5(ΔNEDD8) and HA-CUL5 (ΔRBX) variants (Figure 3B). In addition, HA immunoprecipitations of HA-CUL5(ΔNEDD8) or HA-CUL5(ΔRBX) showed defects in their capability to bind TRIAD1, suggesting a specific requirement for NEDD8 in the interaction of TRIAD1 with CUL5 complex (Figure 3B). To test for a direct physical interaction of TRIAD1 with CUL5-N8, we reconstituted this interaction with a purified, recombinant CUL5–RBX2 complex and MBP-TRIAD1. Recombinant CUL5–RBX2 could be efficiently (~90%) neddylated in a reaction assay containing NEDD8 activating enzyme, the RBX2-specific NEDD8 E2, UBE2F, and NEDD8 (Supplementary Figure S3A). Moreover, the neddylated CUL5–RBX2 complex could catalyse ubiquitin chain formation when used together with the E2 CDC34 (but not with UBCH7) indicating a functional NEDD8-CUL5–RBX2 ligase complex *in vitro* (Supplementary Figure S3B). Non-neddylated and neddylated CUL5–RBX2 complexes were then incubated without or with MBP-TRIAD1. Capture of MBP-TRIAD1 complexes on amylose resin preferentially co-precipitated neddylated CUL5–RBX2 (lane 8) rather than the corresponding non-neddylated complex (lane 6) (Figure 3C). Moreover, size-exclusion chromatography experiments confirmed that TRIAD1 forms a stable complex with NEDD8-CUL5–RBX2, but not with CUL5–RBX2. This complex corresponds to a 1:1 stoichiometric ratio of TRIAD1:NEDD8-CUL5–RBX2 (Figure 3D). Hence, the ligation of NEDD8 significantly enhances the affinity of CUL5–RBX2 for TRIAD1.

This intriguing observation led us to hypothesize the existence of an NEDD8 binding motif within TRIAD1. Indeed, computational analysis predicted a ubiquitin associated (UBA)-like domain preceding the RING1 domain, which is conserved within the Ariadne subfamily of RBR E3 ligases. More recently, the crystal structure of human HHARI confirmed the presence of just such a UBA-like domain (Duda *et al*, 2013), but at that time the significant homology to UBA and available structural data on UBA domains allowed us to predict potential amino acids that could assemble a binding surface for NEDD8 but were most likely not required to maintain the overall structure of the domain (Supplementary Figure S3C). To test the functionality of this UBA-like domain, we generated cell lines expressing GFP-TRIAD1 with amino-acid mutations in the UBA-like domain. The triple mutation GFP-TRIAD1 (L84A, V86A, W100A) and, to a lesser extent GFP-TRIAD1 (W100A), reduced binding to CUL5 indicating a requirement for these amino acids in binding the neddylated cullin (Figure 3E). In addition, we wished to test the possibility of a direct interaction between

MBP-TRIAD1 and NEDD8 in an *in vitro* binding assay. Although we could not observe co-migration of free NEDD8 with TRIAD1 by size-exclusion chromatography, we found that TRIAD1 did bind to NEDD8-coupled agarose beads but not to ubiquitin-coupled agarose beads, suggesting a direct but weak NEDD8/TRIAD1 interaction (Figure 3F). In order to confirm that these beads have an equivalent binding activity we tested, under similar conditions, the binding of the UBA and UIM domain-containing protein UBXN7. In agreement with a recent report, UBXN7 bound with equal efficiency to NEDD8 and ubiquitin (Figure 3F) (Bandau *et al*, 2012). Taken together, these data show that TRIAD1 preferentially interacts with neddylated CUL5 *in vitro* and *in vivo*.

### Neddylated CUL5–RBX2 stimulates TRIAD1 E3 ligase activity

Having determined the physical interaction between TRIAD1 and neddylated CUL5–RBX2 (NEDD8-CUL5–RBX2), we tested whether this interaction has any effect on TRIAD1 ubiquitin ligase activity, assayed by effects on TRIAD1-dependent loss of UBCH7~ubiquitin intermediate. Whereas NEDD8 alone had no effect on the reaction (lane 7), and CUL5–RBX2 (lane 4) had only a minor effect, NEDD8-CUL5–RBX2 strikingly stimulated TRIAD1-dependent disappearance of the UBCH7~ubiquitin intermediate (Figure 4A and B). Importantly, the 130 N-terminal residues in TRIAD1, that are essential for NEDD8-CUL5–RBX2 binding, were also required for enhanced activity in the presence of NEDD8-CUL5–RBX2 (Figure 4C). We also compared TRIAD1 ubiquitylation activity in the presence or absence of either CUL5–RBX2 or NEDD8-CUL5–RBX2. In agreement with the above data, NEDD8-CUL5–RBX2 stimulated TRIAD1 ubiquitylation activity, monitored and quantified as ubiquitin conjugation in immunoblot analyses (Figure 4D and E). Note that UBCH7 serves only as an E2 for TRIAD1 and not for CRLs during such *in vitro* ubiquitylation reactions (Supplementary Figure S3B) (Wenzel *et al*, 2011). Further detailed immunoblot analyses revealed that NEDD8-CUL5–RBX2 enhanced TRIAD1 auto-ubiquitylation, and we also noted that TRIAD1 targets CUL5 for ubiquitylation (Figure 4F and G).

### HHARI binds neddylated cullin-1, -2, -3, -4A complexes

Given the homology within the Ariadne family of RBR E3 ligases, we considered whether the interactions with CRLs might also be conserved. We thus tested the subfamily member HHARI (Ariadne-1, ARIH1) through precipitation of GFP-tagged HHARI from cell lysates, and examination for co-precipitating CRLs by immunoblotting. Strikingly, GFP-HHARI interacted with the neddylated form of CUL1, -2, -3 and -4A but not with CUL4B or CUL5 (Figure 5A; Supplementary Figure S4). Moreover, the interaction with CUL1 was abolished upon pre-treatment of cells with MLN4924. Additional parallels between the TRIAD1/ NEDD8-CUL5–RBX2 and HHARI/NEDD8-CUL1–RBX1 complexes became apparent upon mutation of the conserved acidic or the UBA-like domains at the N-terminus of HHARI (Figure 2E; Supplementary Figure S3C). Precipitation of GFP-HHARI with either an N-terminal 100-aa deletion (GFP-HHARI (aa 101–557)) or triple mutations in the predicted UBA-like domain (V123A, I124A, and W140A) abolished or significantly reduced the interaction with CUL1-NEDD8 (Figure 5B and C), whereas the RING1 H205A mutation

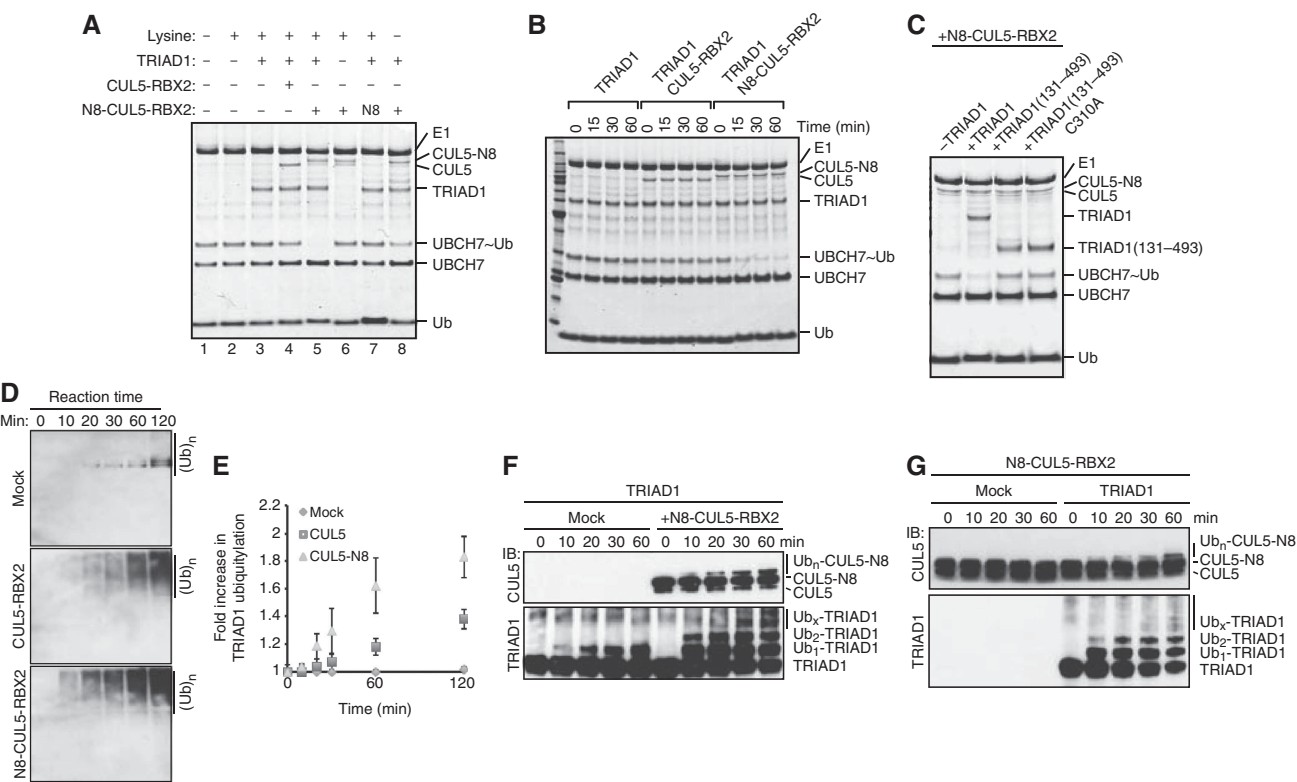

**Figure 4** Neddylated CUL5–RBX2 complex stimulates TRIAD1 E3 ligase activity in vitro. (**A**) UBCH7∼Ub discharge assays with 0.25 μM TRIAD1 were performed in the absence or presence of 30 nM CUL5–RBX2 (lane 4), 30 nM neddylated CUL5–RBX2 (N8-CUL5–RBX2) (lane 5) or free NEDD8 (lane 7). Reaction products were resolved on SDS–PAGE and visualized by SimplyBlue. (**B**) SimplyBlue-stained gel showing time course of UBCH7∼Ub discharge by TRIAD1 in the absence or presence of either 30 nM CUL5–RBX2 or 30 nM N8-CUL5–RBX2. (**C**) UBCH7∼Ub discharge reactions involving 0.25 μM of the indicated TRIAD1 variants were performed in the presence of 30 nM N8-CUL5–RBX2 and reaction products were analysed as described in (**A**). (**D**) Ubiquitylation reactions with UBCH7 and 0.25 μM TRIAD1 were performed in the absence (mock, top panel) or presence of either 30 nM CUL5–RBX2 (middle panel) or 30 nM neddylated CUL5–RBX2 (bottom panel) and measured at indicated time points with anti-ubiquitin antibody. (**E**) Quantification of ubiquitin conjugation as shown in (**D**) was obtained from chemiluminescent immunoblots using ImageJ software analysis. Standard errors of the mean are given from two independent experiments. (**F**) Ubiquitylation reactions with UBCH7 and 0.3 μM TRIAD1 were performed in the absence (mock) or presence of 100 nM N8-CUL5–RBX2 for the indicated times. TRIAD1 auto-ubiquitylation and CUL5 ubiquitylation were analysed by immunoblot analyses. (**G**) Ubiquitylation reactions with UBCH7 and a fixed concentration of 100 nM N8-CUL5–RBX2 were performed in the absence (mock) or presence of 0.3 μM TRIAD1 for the indicated times. TRIAD1 auto-ubiquitylation and CUL5 ubiquitylation were analysed by immunoblot analysis. $Ub_x$-TRIAD1, ubiquitylated TRIAD1; $Ub_n$-CUL5, ubiquitylated CUL5.

significantly decreased UBCH7 binding without disrupting the interaction with CUL1-NEDD8. Furthermore, a direct physical interaction was observed by co-migration of purified HHARI and NEDD8-CUL1–RBX1 using size-exclusion chromatography (Figure 5D). Thus, we conclude that a common feature of at least some Ariadne RBR E3s is their interaction with distinct neddylated CRLs.

### Neddylated CUL1–RBX1 activates auto-inhibited HHARI

The recent finding that HHARI is auto-inhibited (Duda et al, 2013), together with our observation that NEDD8-CUL5–RBX2 stimulates the intrinsic ubiquitin ligase activity of TRIAD1 (Figure 4), led us to investigate whether NEDD8-CUL1–RBX1 might stimulate HHARI E3 ligase activity. We studied HHARI E3 ligase activities in the presence of NEDD8-CUL1–RBX1 complex by utilizing the split and co-expressed form of CUL1–RBX1 (Zheng et al, 2002; Duda et al, 2008). HHARI alone showed only modest auto-ubiquitylation using in vitro reactions with recombinant UBCH7, which serves only as an E2 for HHARI and not for CRLs (Wenzel et al, 2011). However, activity was substantially enhanced in the

presence of NEDD8-CUL1–RBX1 to the extent that rapid HHARI auto-ubiquitylation, even in the presence of excess lysine, precluded observation of UBCH7∼ubiquitin discharge (Figure 6A and C and data not shown). Both the N- and C-terminal domains of CUL1 were also ubiquitylated in these reactions, in a manner dependent on HHARI (Figure 6A). Notably, the neddylated form of the C-terminal domain (CTD) of CUL1 was sufficient to enhance HHARI E3 ligase activity (lane 4) whereas, as observed for TRIAD1, NEDD8 on its own did not enhance HHARI auto-ubiquitylation (lane 6) (Figure 6B and C).

The recently determined crystal structure of HHARI revealed an auto-inhibition by the C-terminal Ariadne domain, masking the RING2 and blocking access to HHARI's catalytic cysteine (Duda et al, 2013). Furthermore, the HHARI RING2, like the corresponding domain from PARKIN, displays an arrangement of side chains resembling a catalytic triad from cysteine proteases (Wenzel et al, 2011; Duda et al, 2013; Riley et al, 2013; Spratt et al, 2013; Trempe et al, 2013; Wauer and Komander, 2013). Accordingly, PARKIN was shown to react with chemically modified versions of ubiquitin harbouring

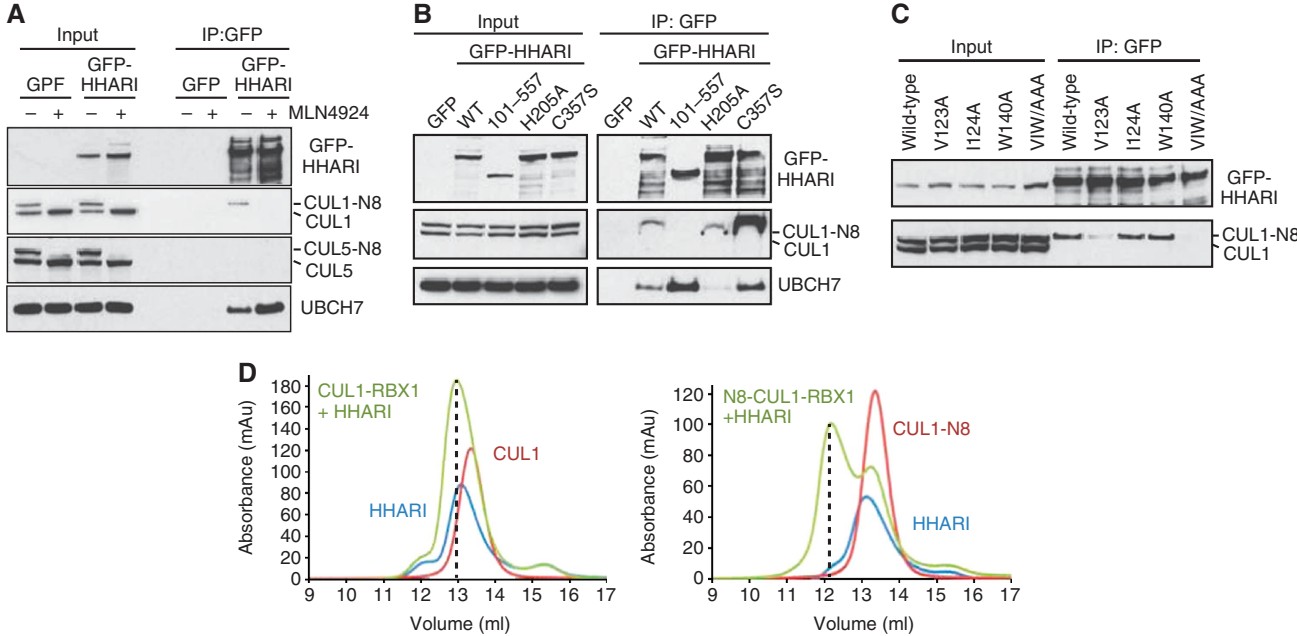

**Figure 5** HHARI binds neddylated cullin-1 complexes. (**A**) HEK293 cell lines stably expressing GFP or GFP-HHARI were either mock ($-$) treated or treated overnight with 1 µM MLN4924 ($+$) before immunoprecipitation with anti-GFP antibodies and subsequent immunoblotting with the indicated antibodies. (**B**, **C**) Stably expressed GFP-tagged WT, N-terminally truncated (aa 101–557) and mutated HHARI (as indicated) were immunoprecipitated from HEK293 cells with anti-GFP antibodies and immunoblotted to detect CUL1 and UBCH7 binding. VIW/AAA refers to a V123A/I124A/W140A mutant. (**D**) Gel filtration chromatography analysis of recombinant HHARI mixed either with non-neddylated (left graph) or with neddylated CUL1–RBX1 (right graph).

C-terminal electrophiles, which have largely been developed for reactivity with catalytic triad-based deubiquitylating enzymes (Riley *et al*, 2013; Wauer and Komander, 2013). We found that full-length, auto-inhibited HHARI did not react with the electrophilic Ubiquitin-vinyl methyl ester (Ub-VME), whereas a mutant version of HHARI that is artificially activated through deletion of the inhibitory Ariadne domain (ΔARI) (Duda *et al*, 2013) did react with Ub-VME (Figure 6D and E). As observed previously for intrinsic auto-ubiquityla-tion activity (Duda *et al*, 2013), the reactivity with Ub-VME was partially blocked by adding a molar excess of the isolated Ariadne domain (HHARI[ARI]) but not with a structure-based mutant version (F430A, E431A, E503A) (Figure 6E) (Duda *et al*, 2013). Strikingly, NEDD8-CUL1–RBX1 greatly stimulates HHARI reactivity with Ub-VME, to an extent similar to that observed for HHARI(ΔARI). In contrast, CUL1–RBX1 and NEDD8 had only a minor or no effect (Figure 6D). Analogous experiments revealed that NEDD8-CUL5–RBX2 likewise stimulated the reaction between TRIAD1 and Ub-VME, although we were unable to produce TRIAD1 protein lacking the Ariadne domain to unambiguously determine its role in TRIAD1 (Figure 6F). Taken together, these data suggest that interactions with cognate NEDD8-CRLs cause a conformational change that exposes the RING2 catalytic cysteine in the Ariadne RBR E3 partner.

### TRIAD1 and HHARI impact on cullin neddylation in vivo
Given the stable interactions that we observed for TRIAD1 and HHARI with neddylated cullins, we asked whether TRIAD1 and HHARI might affect CRL neddylation in cells. We monitored CUL5-NEDD8:CUL5 ratios in cells expressing either wild-type or mutant versions of GFP-TRIAD1. Interestingly, cells expressing the RING2 mutant GFP-

TRIAD1(C300A) showed a significant increase in CUL5-NEDD8 protein levels that also resulted in an increased amount of co-precipitated CUL5-NEDD8 in GFP-TRIAD1 im-munoprecipitates (Figure 7A). Similar results were obtained for mutations in TRIAD1 zinc-binding residues Cys300 and Cys318 that would disturb the structural integrity of RING2, in mutations of the putative active site cysteine residue Cys310 (Figure 7B), or in a *TRIAD1* deletion in avian DT40 cells complemented with RING2-mutated (C300A) human TRIAD1 (clones C and D) (Supplementary Figure S5C).

We further investigated whether the elevated CUL5-NEDD8:CUL5 ratios in TRIAD1(C310S) expressing cells were due to altered neddylation/de-neddylation rates using the MLN4924 inhibitor. HEK293 cells were treated with MLN4924 over different periods of time and cullin de-neddy-lation was monitored by immunoblotting (Figure 7C). GFP control and GFP-TRIAD1 expressing cells were efficiently de-neddylated after 15 min. In contrast, GFP-TRIAD1(C310S) expressing cells displayed an overall higher level of neddy-lated CUL5, and retained significant levels of CUL5-NEDD8 (but not of other cullins such as CUL2) after 60 min of treatment (Figure 7C). To test for re-neddylation, we treated cells for 3 h with MLN4924 to completely abolish the neddy-lated form of cullins ($t = 0$), washed out the drug and allowed cells to recover (Supplementary Figure S5D). GFP and GFP-TRIAD1 expressing cells only partially recovered their steady-state CUL5-NEDD8:CUL5 ratios after 300 min, whereas GFP-TRIAD1(C310S) expressing cells completely recovered their CUL5-NEDD8:CUL5 ratios to levels observed in untreated control.

We next asked whether the expression of a C357S RING2 mutant of HHARI might have a similar effect on CUL1. Indeed, we observe an increased CUL1-NEDD8:CUL1 ratio

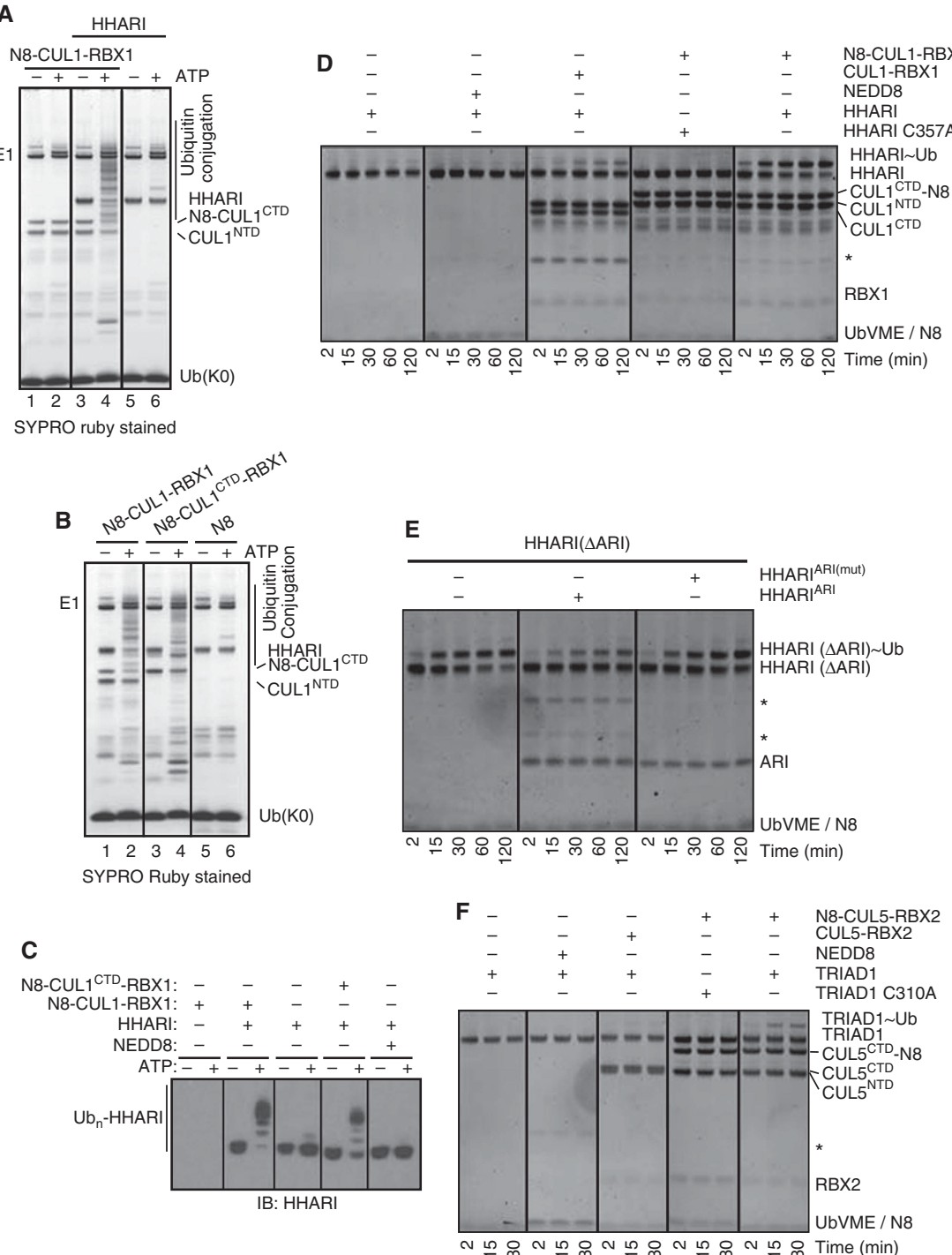

**Figure 6** Neddylated CUL1–RBX1 activates auto-inhibited HHARI. (**A**, **B**) Ubiquitylation reactions were carried out ±ATP with UBCH7 and non-conjugatable lysine-less ubiquitin in the absence or presence of HHARI, N8-CUL1–RBX1, N8-CUL1$^{CTD}$–RBX1, and NEDD8 as indicated. Reaction products were visualized by SYPRO Ruby staining. (**C**) Ubiquitylation reactions were carried out as in (**A**) and (**B**) and reaction products were subjected to immunoblot analysis with HHARI antibody. (**D**) SYPRO Ruby-stained gel showing time-dependent covalent modification of HHARI with Ub-vinyl methyl ester (Ub-VME) in the absence or presence of CUL1–RBX1, N8-CUL1–RBX1, and NEDD8. (**E**) Covalent modification of HHARI (ΔAri) with Ub-VME in the absence (−) or presence (+) of either isolated wild-type or mutated (F430A, E431A, E503A) Ariadne domain over indicated time. Reaction products were visualized by SYPRO Ruby staining. (**F**) SYPRO Ruby-stained gel showing time-dependent formation of TRIAD1~Ub in the absence or presence of CUL5–RBX2, N8-CUL5–RBX2, and NEDD8. *Unspecific bands.

and CUL1-NEDD8 precipitation, but no effect on CUL5-NEDD8:CUL5 in GFP-HHARI(C357S) cells (Figure 7D). Using a de-neddylation experiment like that described in Figure 7C we observed a similar, but more modest, delay in

CUL1 de-neddylation kinetics in the presence of GFP-HHARI (C357S) (Figure 7E). To further assess the effect of an altered CUL1 neddylation pattern on CUL1 complex activity, we analysed the abundance of proteins that are subject to

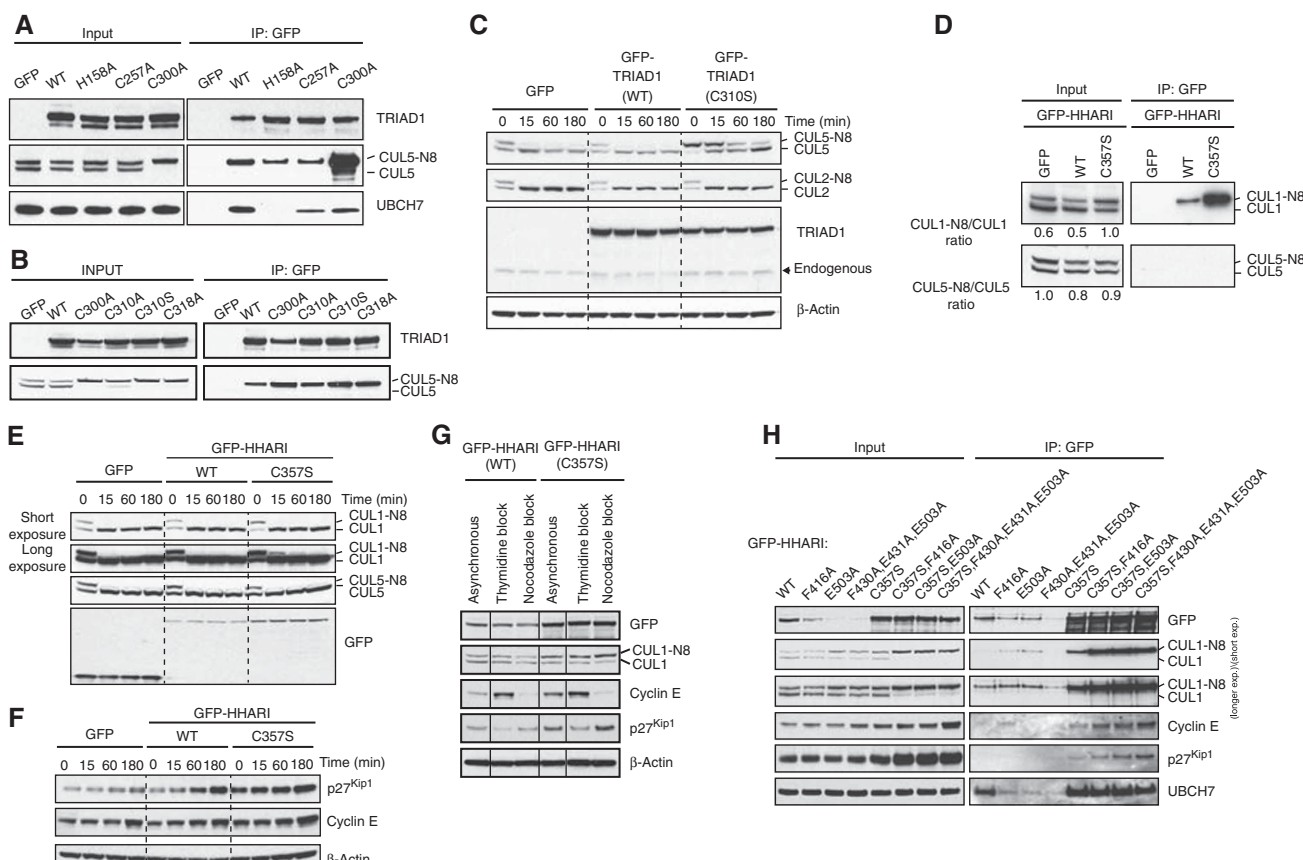

**Figure 7** Dominant-negative mutants of TRIAD1 and HHARI impact on cullin neddylation *in vivo*. (**A**, **B**) GFP, or GFP-tagged WT or mutant TRIAD1 containing the indicated amino-acid substitutions was stably expressed in HEK293 cells and immunoprecipitated using anti-GFP agarose. (**C**) HEK293 cell lines expressing GFP, GFP-TRIAD1, or RING2-mutated GFP-TRIAD1(C310S) were treated with 3 μM MLN4924 inhibitor over the indicated time periods and subjected to immunoblot analyses. (**D**) GFP-immunoprecipitations were performed from HEK293 cell lines stably expressing GFP, GFP-HHARI, or RING2-mutated GFP-HHARI(C357S). Samples were analysed for cullin neddylation by immunoblotting. The ratios of neddylated to unmodified cullins were obtained from these autoradiograms using ImageJ analysis software. (**E**, **F**) HEK293 cell lines expressing GFP, GFP-HHARI, or GFP-HHARI(C357S) were treated with 0.3 μM MLN4924 for the indicated time periods and subjected to immunoblot analyses. (**G**) HEK293 cells expressing GFP-HHARI (wild-type or C357S mutant) were treated with thymidine or nocodazole, or left untreated (asynchronous) and subjected to immunoblot analyses. (**H**) The indicated GFP-HHARI constructs were expressed in HEK293 cells and subjected to immunoprecipitation with anti-GFP antibodies before immunoblot analysis with the indicated antibodies.

CUL1-mediated degradation such as p27[Kip1] and cyclin E. During the course of a de-neddylation experiment in cells expressing GFP and GFP-HHARI, we observed accumulation of endogeneous p27[Kip1] and cyclin E in accordance with the de-neddylation and deactivation of the CUL1 ligase complex (Figure 7F). However, those cells expressing GFP-HHARI(C357S) showed an enhanced accumulation of endogeneous p27[Kip1] and cyclin E, indicating an overall reduced CUL1 ligase activity in the presence of this mutant. In addition, we treated cells by thymidine or nocodazole block. Cells expressing the GFP-HHARI(C357S) mutant accumulated cyclin E following thymidine block and p27[Kip1] accumulated after nocodazole block suggesting a defect in CUL1-mediated degradation of p27[Kip1] and cyclin E during certain cell-cycle phases (Figure 7G; Supplementary Figure S5E).

We next tested the effects of mutations in the HHARI Ariadne domain, which were recently shown to relieve auto-inhibition and result in enhanced HHARI auto-ubiquitylation *in vitro* (Duda *et al*, 2013) (Figure 7H). We observed less GFP-HHARI overall, potentially reflecting effects of auto-ubiquitylation, and a concomitant increase in CUL1-NEDD8 levels. Inactivation of E3 ligase activity by means of an

additional catalytic C357S mutation restored accumulation of these Ariadne domain variants, as did treatment with the proteasomal inhibitor MG132 (Supplementary Figure S5F). These data suggest that unregulated auto-ubiquitylation may render HHARI susceptible to proteasomal degradation, as has been speculated for the RBR ligase Parkin (Chaugule *et al*, 2011).

Interestingly, combining the catalytically inactivating C357S mutant with mutants designed to abrogate the intramolecular RING2–Ariadne interaction further enhanced the CUL1-N8:CUL1 ratio, and GFP immunoprecipitations from cells expressing these GFP-HHARI variants (C357S, F416A), (C357S, E503A), and (C357S, F430A, E431A, E503A) resulted in significantly increased amounts of co-precipitated CUL1-NEDD8. In agreement with observations made in Figure 7F and G, these cells also accumulate the CUL1–RBX1 substrates p27[Kip1] and cyclin E (Figure 7H).

## Discussion

Here, we provide evidence that Ariadne RBR E3 ubiquitin ligases such as TRIAD1 and HHARI can bind and be activated

by CRL complexes. Whereas TRIAD1 specifically associates with CUL5–RBX2, HHARI is more promiscuous towards cullin types and associates with RBX1-associated cullins 1, 2, 3, and 4A. Interestingly, both TRIAD1 and HHARI show a strong preference for binding the neddylated form of the cullin. Our data suggest a novel function of NEDD8 in directing specific CRLs to Ariadne RBR ligases, which in turn exert influence on the levels of their cognate neddylated cullin.

A general feature that has emerged from recent studies is that RBR E3 ligases, such as HOIP, PARKIN, and HHARI, are auto-inhibited (Chaugule et al, 2011; Kondapalli et al, 2012; Smit et al, 2012; Stieglitz et al, 2012; Duda et al, 2013; Riley et al, 2013; Spratt et al, 2013; Trempe et al, 2013; Wauer and Komander, 2013). At present, though, we are only beginning to understand the mechanisms underlying how auto-inhibition is relieved. The current understanding of HOIP activation is that the UBL domain of HOIL-1L or SHARPIN interacts with HOIP's UBA domain to release auto-inhibitory domains, which have not yet been identified, to promote linear ubiquitin chain formation (Smit et al, 2012; Stieglitz et al, 2012). The structure of the auto-inhibited form of PARKIN uncovered that access to the catalytic Cys431 is blocked by another domain, and in addition, that a linker helix (referred to as the repressor element of PARKIN (REP)) binds and occludes the E2 binding face of RING1 (Riley et al, 2013; Trempe et al, 2013; Wauer and Komander, 2013). Furthermore, the auto-inhibited form of the ligase reveals that the active site Cys and the E2 active site are separated by an insurmountable >50 Å gap. One potential mode of stimulating PARKIN's activity is by PINK1-dependent phosphorylation of Ser65 in the UBL domain and the release of the auto-inhibitory effect (Kondapalli et al, 2012). Recently, the crystal structure of full-length human HHARI revealed a distinct but related mechanism for HHARI's auto-inhibition (Duda et al, 2013). The C-terminal Ariadne domain masks the catalytic Cys357, but the positioning of the Ariadne domain also maintains a wide separation between the predicted active site of RING1-bound E2 and Cys357, hence making Cys357 inaccessible for efficient ubiquitin trans-thiolation (Figure 8). Given the domain similarities, in particular the Ariadne domain, other members of the Ariadne RBR subfamily may be auto-inhibited in a similar fashion.

We propose a model whereby binding of the Ariadne RBR E3 ligase to neddylated cullins relieves auto-inhibition (Figure 8). We speculate that the activated conformation is stabilized through multisite interactions: The acidic domain of the Ariadne RBR E3 may interact with the basic canyon within the cullin; the Ariadne E3's UBA-like domain might bind NEDD8; and the Ariadne domain may be displaced from the RING2 active site through additional interactions with the neddylated cullin. Irrespective of the structural mechanism, our data demonstrate that neddylated cullins lead to exposure and reactivity of the RING2 catalytic cysteine (Figure 6D–F).

It has become increasingly clear that NEDD8 plays numerous roles in altering the activities of CRLs. Aside from its well-established role as a multimodal activator of CRL complex activity (Saha and Deshaies, 2008; Duda et al, 2011), NEDD8 also plays a pivotal role in governing the exchange of F-box protein containing substrate receptors via its antagonism of CAND1 binding (Pierce et al, 2013; Wu et al, 2013; Zemla et al, 2013; and additional references therein), as

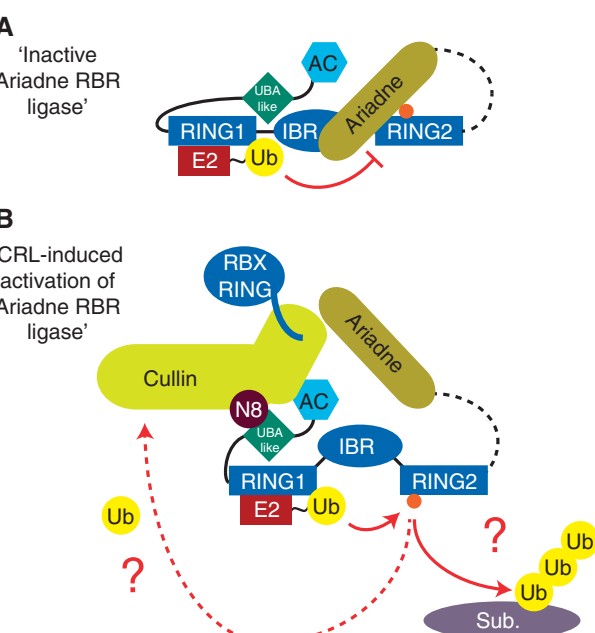

**Figure 8** Model for CRL-induced activation of Ariadne RBR ligases. (**A**) Auto-inhibited Ariadne RBR ligase based on the HHARI structure (Duda et al, 2013). (**B**) Illustration of the possible activation of the Ariadne RBR ligase by neddylated cullin as described in Discussion. Orange ball, catalytic Cys; AC, acidic domain; UBA-like, ubiquitin-association (UBA)-like domain.

well as interacting with additional protein partners to direct CRLs to other machineries involved in substrate degradation, such as the p97 segregase (Bandau et al, 2012; den Besten et al, 2012). Here, we show that NEDD8 is also important for targeting specific CRLs to members of the Ariadne subfamily of RBR E3 ligases.

Not only is Ariadne RBR ligase activity stimulated by CRL complex binding in vitro, but a reciprocal functional relationship also seems to exist in vivo, impinging on cullin neddylation. Cells expressing versions of TRIAD1 and HHARI mutated in their RING2 domains exhibited increased steady-state levels of neddylated cullin and protected those CRLs from de-neddylation in the presence of MLN4924, although future studies will be required to determine the mechanisms by which TRIAD1 and HHARI impact de-neddylation. In addition, we observed an accumulation of the CUL1–RBX1 substrates p27[Kip1] and cyclin E in HHARI(C357S) expressing cells, leading us to conclude that this neddylated CUL1 complex is functionally less active. It remains to be resolved mechanistically how Ariadne RBR ligase binding and activity affects CRLs, but it is possible that the Ariadne RBR ligase targets CRL complex components or exerts activity-dependent steric effects that alter neddylation (Figure 8). We have also provided evidence that constitutively active HHARI, and likely CRL-activated HHARI, promotes its own proteasomal degradation by auto-ubiquitylation. Hence, auto-inhibition might be a mechanism employed by Ariadne RBR ligases to avoid misdirected auto-ubiquitylation and consequent degradation.

The discovery and initial bioinformatic characterization of the RBR E3 ligase family suggested that some RBRs might be assembled into multiprotein ubiquitin ligase complexes (Marin and Ferrus, 2002; Marin et al, 2004). One member

of the Ariadne subfamily, the atypical cullin protein CUL9 (PARC) has, in addition to the RBR domain, a characteristic cullin domain that is closely related to the cullin CUL7. This CUL7 domain has been shown to bind the RING E3 ligase RBX1 and to be neddylated, suggesting that CUL9 forms CRLs (Skaar et al, 2007). It is well established that gene fusions tend to occur between genes that encode functionally related, or physically interacting, proteins ('Rosetta Stone strategy') and it was demonstrated that *CUL9* originates from an ancient gene fusion between an *Ariadne* gene and a duplication of the *CUL7* gene (Marcotte et al, 1999; Marin and Ferrus, 2002; Marin et al, 2004). Here, our studies provide new experimental evidence for RBR/cullin complexes by showing that two members of the Ariadne subfamily are physically and functionally assembled in CRL complexes. This strongly indicates that protein complex formation might be a common feature of Ariadne RBR ligases and that CUL9 is indeed an ancient prototype of Ariadne/cullin complexes.

In summary, our studies provide fundamental insights into the regulatory mechanisms by which Ariadne RBR ligase auto-inhibition is relieved, while at the same time revealing a novel role of NEDD8 in recruiting additional E3 ligases to existing CRL E3 ubiquitin ligase complexes. We envison that our work will form the foundation of further structural/functional studies in the near future to determine the molecular mechanisms underlying the inter-dependent regulation of Ariadne RBR/CRL complexes.

## Materials and methods

### Protein expression and purification

NEDD8 was expressed in *E. coli* and purified from inclusion bodies (Whitby et al, 1998). Expression of the NEDD8-specific E1 enzyme took advantage of the newly described Dac tag (Lee et al, 2012). Briefly, Dac-TEV-APPBP1/UBA3 heterodimer was expressed in Sf21 insect cells then lysed in 30 mM HEPES pH 7.5, 0.2% Triton X-100, 0.5 mM EDTA, 0.5 mM EGTA, 1 mM TCEP, 1 mM Pefabloc, 20 μg/μl Leupeptin. Insoluble material was removed by centrifugation and the lysate was incubated with ampicillin Sepharose. The Sepharose was collected by centrifugation, washed, and proteins were eluted with 30 mM HEPES, 5% glycerol, 150 mM NaCl, 10 mM ampicillin, 1 mM TCEP, 0.03% Brij-35. The Dac tag was removed by TEV-protease treatment. Dac-TEV-Cullin-5–RBX2 heterodimer was expressed and purified in a similar way. GST-UBE2T was expressed in bacteria and purified on glutathione-sepharose as described elsewhere (Alpi et al, 2008). UBE2F, UBE2M, CDC34, and UBCH7 were expressed as His-tagged proteins in BL21 cells and purified by $Ni^{2+}$-Sepharose chromatography. GST-TRIAD1 (and the various mutant versions described in the text) was also expressed in BL21 cells and purified by GSH-Sepharose chromatography in the presence of 2 μM $ZnCl_2$. Tag-free protein was generated by cleavage with PreScission protease. MBP-TRIAD1-$His_6$ was first purified on amylose resin then by $Ni^{2+}$-Sepharose chromatography. CUL1–RBX1 complex was expressed and purified as described (Duda et al, 2008). HHARI was purified by glutathione affinity chromatography as described elsewhere (Duda et al, 2013).

### Neddylation and ubiquitylation assays

Neddylation reactions contained 30 nM NEDD8 E1, 40 nM E2, 10 μM NEDD8 and 0.5 μM purified recombinant CUL5–RBX2 in 50 mM HEPES pH 7.4, 5 mM $MgCl_2$, 60 mM KOAc, 10% glycerol, 1 mM DTT, 0.4 mM EDTA, and 5 mM $Mg^{2+}$-ATP. Reactions were incubated for 60 min at 37°C. Ubiquitylation reactions contained 80 nM E1, 0.60 μM E2, 25 μM ubiquitin, and indicated concentrations of relevant E3 ubiquitin ligase in 1 × PBS. Reactions were stopped by adding SDS sample buffer supplemented with 5 mM 2-mercaptoethanol, and boiled for 5 min. Reaction products were analysed by immunoblot-

ting. E2 screening was carried out using Ubiquigent's $E2^{scan}$ kit according to the manufacturer's instructions.

### UBCH7~ubiquitin discharge and Ub-VME binding assays

UBCH7 was first charged with ubiquitin in the absence of TRIAD1. To generate the UBCH7~Ub thioester, 10 μM of ubiquitin and 3 μM of UBCH7 were incubated with 4 μM Ube1 in 50 mM HEPES pH 7.5, 2 mM ATP and 2 mM $MgCl_2$ at 37°C for 60 min. E1-mediated loading of UBCH7 was stopped by depleting ATP with 0.5 U/ml apyrase. The efficiency of UBCH7~Ub formation varied from 30 to 50%. In all, 10 μl of UBCH7~Ub reaction was incubated with TRIAD1 (0–3.6 μM, as indicated in figure legends) and 22 mM lysine in a final volume of 20 μl at 37°C for various times, as indicated in figure legends. Reactions were terminated by addition of non-reducing SDS–PAGE sample buffer. Reaction products were resolved on 4–12% Bis/Tris SDS–PAGE gels and stained either with SimplyBlue SafeStain or subjected to immunoblot analysis. The modification of TRIAD1 and HHARI with Ub-VME suicide probes was performed according to recently described experimental procedures (Wauer and Komander, 2013).

### TRIAD1/UBCH7 and TRIAD1/CUL5 in vitro binding assays

In all, 10 μg of $His_6$-TRIAD1 was mixed with 2 μg of untagged UBCH7 in Binding buffer (50 mM Tris–HCl pH 8.0, 10 μM $ZnCl_2$, 600 mM NaCl, 10% glycerol, 0.5% NP-40, 5 mM 2-mercaptoethanol) and incubated at 4°C. $His_6$-TRIAD1/UBCH7 complexes were precipitated by $Ni^{2+}$-Sepharose affinity chromatography and precipitates were resolved on SDS–PAGE followed by immunoblot analysis. In all, 1.5 μg of recombinant CUL5–RBX2 was either neddylated or 'mock neddylated' in the absence of UBE2F in reaction buffer (50 mM HEPES-NaOH pH 7.4, 5 mM $MgCl_2$, 60 mM KOAc, 10% glycerol, 1 mM DTT, 0.4 mM EDTA and 5 mM $Mg^{2+}$-ATP) for 60 min at 37°C. Subsequently, 10 μg of recombinant MBP-TRIAD1 was added and the reactions were incubated on ice. Potential MBP-TRIAD1/CUL5–RBX2 complexes were precipitated with amylose resin and washed with reaction buffer. Reaction supernatants (non-precipitated) and precipitates were resolved by SDS–PAGE and subjected to immunoblot analysis.

### Bioinformatic identification of the UBA-like domain

The UBA-like domain was recognized as a region of localized similarity between mammalian TRIAD1, HHARI, and several RBR ligases from different organisms. A multiple alignment of the conserved region was calculated by the MAFFT alignment program, using the L-INS-I algorithm (Katoh et al, 2002). A significant similarity of this region was established by running HHPRED against a collection of alignments assembled from Pfam and other domain alignments (KH, unpublished) (Soding et al, 2005; Punta et al, 2012). The best database match was the UBA domain of the DCN1 family of NEDD8 ligases at a P-value of 2.5E−7 (Kurz et al, 2008), followed by several other UBA families with P-values better than 1E–5.

### NEDD8-binding assay

Binding of TRIAD1 to NEDD8 agarose was carried out as described previously (Kurz et al, 2005).

### Cell extracts, immunoprecipitation, and immunoblot analysis

Stably transfected Flp-In T-Rex-293 cells were stimulated with 1 μg/ml tetracycline overnight to induce expression of the desired fusion proteins. Cells were treated with MLN4924 (Active Biochem) as described in the appropriate figure legends. Where indicated, cells were treated with MG132 at a final concentration of 20 μM for 3 h prior to lysis. Whole-cell extracts were prepared from mammalian cells by lysis in 40 mM HEPES pH 7.4, 120 mM NaCl, 1 mM EDTA, 1% Triton X-100, 2.5 mg/ml dithiobis(succinimidylpropionate) and 'Complete' protease inhibitor cocktail (Roche) for 30 min on ice, before clarification by centrifugation. Whole-cell lysates were prepared from DT40 cells as described previously (Kelsall et al, 2012). GFP-tagged proteins were isolated by immunoprecipitation from 3 mg of cell lysate using 30 μl GFP-Trap agarose beads (Chromotek) incubated for 1 h at 4°C. HA-tagged proteins were captured from 3 mg total cell lysate using anti-HA affinity matrix for 1 h at 4°C. For immunoprecipitation of endogeneous proteins, 50 μg of antibody was incubated overnight with 5 mg of cell lysate at 4°C. In all, 50 μl of Protein-G agarose was added and the reaction was incubated for

a further 2 h. All immunoprecipitation reactions were washed in lysis buffer to remove non-specific binding before elution of the immunoadsorbed proteins by boiling in reducing SDS sample buffer. For immunoblotting, protein samples were separated by SDS–PAGE and transferred onto nitrocellulose membrane. Primary antibodies were used at a concentration of 0.5–1.0 µg/ml or at dilutions recommended by the manufacturers.

### Identification of TRIAD1-interacting proteins by mass spectrometry

GFP-TRIAD1 protein was purified from cells using GFP-Trap agarose beads, as detailed earlier. The immunoprecipitated proteins were separated by SDS–PAGE and Coomassie stained. The gel was divided into 11 gel slices per lane and the proteins within each gel slice were alkylated and digested with trypsin as described elsewhere (Shevchenko *et al*, 2006). The resultant peptides were analysed by means of liquid chromatography-mass spectrometry on a Thermo LTQ-Orbitrap.

### Cell-cycle study

Stably transfected HEK293 cells were treated with 2 mM thymidine or 100 ng/ml nocodazole for 14 h. Cell-cycle profiles were analysed by flow cytometry using propidium iodide staining to assess cellular DNA content.

Details of antibodies used, the disruption of the chicken ARIH2 gene locus, and the generation of complementation cell lines are found in Supplementary Data.

### Supplementary data

Supplementary data are available at *The EMBO Journal* Online (http://www.embojournal.org).

## Acknowledgements

We gratefully acknowledge the excellent technical support provided by the Mass Spectrometry Service and the Division of Signal Transduction Therapy (DSTT). We are grateful to Francesco Melandri and Boston Biochem for providing Ub-VME activity probe. This work was supported by the Scottish Institute for Cell Signalling and the pharmaceutical companies supporting DSTT (AstraZeneca, Boehringer Ingelheim, GlaxoSmithKline, Janssen Pharmaceutica, Merck-Serono, and Pfizer) and in part by the Wellcome Trust Strategic Award grant 097945/B/11/Z. The work from BAS, DMD, and JLO was supported by NIH R01GM065930, and the Howard Hughes Medical Institute. BAS is an investigator of the Howard Hughes Medical Institute.

*Author contributions*: IRK, DMD, JLO, and FL performed all experiments and analysed the data; KH performed the bioinformatics analyses; AK provided proteins used in the described assays; NW and MW provided molecular biology support; BAS designed and conceived experiments and wrote the manuscript; AFA designed the project, conceived the experiments, and wrote the manuscript.

## Conflict of interest

The authors declare that they have no conflict of interest.

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
