## [Review Process File · The EMBO Journal]

Manuscript EMBO-2013-84802

TRIAD1 and HHARI bind to and are activated by distinct neddylated Cullin RING ligase complexes

Ian R. Kelsall, David M. Duda, Jennifer L. Olszewski, Kay Hofmann, Axel Knebel, Frédéric Langevin, Nicola Wood, Melanie Wightman, Brenda A. Schulman, Arno F. Alpi

Corresponding author: Arno F. Alpi, Scottish Institute for Cell Signalling

Review timeline:

Submission date:	15 February 2013
Editorial Decision:	22 March 2013
Revision received:	22 July 2013
Editorial Decision:	26 August 2013
Revision received:	03 September 2013
Accepted:	04 September 2013

Editor: Hartmut Vodermaier

Transaction Report:

Additional correspondence (editor)

14 March 2013

Thank you again for submitting your manuscript EMBOJ-2013-84802, "TRIAD1 and HHARI bind to and are activated by distinct neddylated Cullin RING ligase complexes" for consideration by The EMBO Journal. We have now received all the reports from three experts, which I am enclosing copied below. As you will see, the reviewers express some interest in the basic finding of your study, yet they also raise a number of substantive concerns especially related to the functional follow-up experiments. Before taking a final decision, I would like to give you the opportunity to consider and respond to these comments, detailing whether and how you might be able to address the concerns if offered the possibility to revise the manuscript. In particular, we would like to hear whether (and how) you could imagine improving the criticized functional studies, or whether you could imagine alternative biochemical or structural follow-up characterizations (as suggested by referee 3) that would allow to de-emphasize the currently less conclusive physiological analyses. These tentative response (parts of which we may choose to share and discuss with some of the referees) would be taken into account when making our final decision on this manuscript. I would therefore appreciate if you could send us such a response at your earliest convenience, ideally by early next week.

REFeree REPORTS:

Referee #1 :

The manuscript by Kellsall et al. aims to provide insight into the function and regulation of members of the RBR ligase family which have recently been shown to form the new family of RING/HECT hybrids that combine the activities of both, RING and HECT-type ligases. The authors make the interesting observation that the RBR ligases TRIAD1 and HHARI associate with neddylated cullin-RING ligase complexes and that this interaction positively modifies their ligase activity as judged by auto-ubiquitination assays. Furthermore, they map the region in the RBRs responsible for the interaction to a conserved acidic region and an UBA-like domain that specifically recognize NEDD8 and provide evidence that the RBR ligase activity in turn regulates CRL activity.

This is a very interesting story that identifies yet another level of regulation within the ubiquitination system and should be of great interest to a broad audience. However, the authors should address a few issues to make their manuscript easier accessible and strengthen their model.

Major points

1- To identify cognate E2s for TRIAD1 the authors carried out IPs of endogenous TRIAD1. Not all E3s form stable complexes with their cognate E2s and hence the authors may have missed physiologically important E2s with this approach. It would be good if they added a comment explaining why they didn't try other approaches and if they think there might be other E2s that work with TRIAD1. More importantly, the authors carry out some of the experiments with UBE2T, others with UbcH7 and some with both E2s. Why not do everything with both? Did they try to detect the thioester using both E2s or only UbcH7? And do they think there's a functional difference between the two E2s? Please comment

In this respect, why didn't they find UBE2T in the IP MS experiment?

2- The authors show that TRIAD1 binds tighter to neddylated CUL5-RBX2 than the non-neddylated version and that TRIAD1 can even bind to isolated NEDD8. Given these observations it would be very interesting to see pull-downs directly comparing the relative affinities of TRIAD1 for NEDD8 versus neddylated CUL5-RBX2, i.e. does CUL5 strengthen the interaction or is the interface dominated by NEDD8? And how does isolated NEDD8 affect auto-ubiquitination? Can it activate on its own?

3- The authors show convincingly that the interaction between TRIAD1 and neddylated CUL5-RBX2 stimulates the E3 ligase activity of TRIAD1. But what's the mechanism behind this? As the authors point out themselves in the Discussion this could be due to auto-inhibitory interactions as shown previously for PARKIN and HOIP. This should be tested in detail. What is the difference, if any, of the auto-ubiquitination activity of TRIAD1 in the presence and absence of the region N-terminal to the RBR? Are there differences between UbcH7 and UBE2T? And how does this compare to HHARI?

Minor points

- Introduction page 4, it would be helpful for the reader if a sentence describing the function of mammalian HHARI was added.

- Introduction: "...it remains to be addressed, how Ariadne RBR ligases can function as E3 ligases." What are the authors trying to say with this sentence? Are they referring to regulation of their activity? - Please be more specific.

- Introduction: "...distinct but specifically neddylated..." What do the authors mean? Could it be anything else than "specifically" neddylated?

- On page 8 the authors mention the "basic canyon" and that more of the Elongin-B/-C adaptor complex co-precipitated with mutants that didn't bind to TRIAD1 anymore. Could you please add one or two explanatory sentences about the basic canyon (maybe a figure in Supplementary) and what the significance of the Elongin-B/-C adaptor complex co-IP is?

- It would be helpful to re-blot the auto-ubiquitination assay blots with the anti-TRIAD1 antibody.

- This might seem very picky, but I'm not convinced that @ is a good symbol for a figure.

- The alignment in Fig 3E would be much easier to understand if the authors used colour to highlight conserved residues.

Referee #2:

The RBR-family of E3s was recently shown to catalyze ubiquitylation via an obligate thioester intermediate, using a Cys residue in its RING2-domain. Although some members of the RBR-family, such as the E3s Parkin or LUBAC, are heavily studied, most RBRs remain rather poorly understood. Determining mechanisms that underlie the function or regulation of RBRs would be important and interesting for a wide readership.

In this manuscript, Kellsall et al. investigated the biochemical function of Triad1 and Hhari, two members of the Ariadne-class of RBRs. Biochemical analysis suggested that Triad1 acts as a canonical RBR, although a thioester-bond between a critical Cys residue in its RING2 and ubiquitin could not be detected. In cells, Triad1 was found to interact specifically with the neddylated form of Cul5, but not with other Cullin-RING ligases. HHARI similarly interacted with neddylated cullins, although it was less specific (with the exception that, interestingly, it did not interact with Cul5). The interactions between Triad1/HHARI and N8-Cul were direct and mediated via a UBA-domain in Triad1 and HHARI. Up to this part, this paper is very nice, and I have only very minor issues, as described below.

However, the authors next analyze the functional consequences of the interaction between Triad1 and N8-Cul5, and I found this part of the paper a bit unsatisfying. They report an increase in the apparent ubiquitylation activity of Triad1, if Cul5 was present. Unfortunately, although Triad1 appears to bind specifically to neddylated Cul5, the *in vitro* assays shown in Fig. 4a reveal only minor differences between unneddylated and neddylated Cul5, raising questions about the specificity of this assay. Raising similar concerns, N8-Cul5 only allows addition of one extra ubiquitin to Triad1 in autoubiquitylation assays - in the absence of a known function of Triad1 autoubiquitylation, it is unclear whether this event is significant. Whether Cul5 really stimulates Triad1 activity requires either an analysis of Triad1-substrate ubiquitylation or, at least, a more in-depth biochemical analysis of the autoubiquitylation or chain formation (i.e. quantitative kinetics).

In contrast to these rather weak effects on Triad1-autoubiquitylation, the authors show that Triad1 catalyzes monoubiquitylation of Cul5, and given their later findings, it is possible that this reaction plays a role in regulating the CRL. Is monoubiquitylation of Cul5 dependent on neddylation? What is the site of this reaction (they have all components reconstituted, so should be able to determine the modification site by mass spectrometry)? Does interfering with this modification affect the activity of Cul5? Because very little is known about physiological substrates of Triad1 or HHARI, analyzing consequences of TRIAD1 on CRL-substrate ubiquitylation or degradation provides a more straightforward means to assessing the physiological importance of the reported interaction (which, given a recent Nature Immun. Paper on Triad1 is clear!). Therefore, experiments describing SCF- or Cul5-substrate modification and stability in the presence of HHARI or Triad1 could greatly strengthen the paper.

Minor issues:

1. Fig. 1B: different exposures between E2s make it almost impossible to judge the specificity of these IPs. Given the unbiased mass spectrometry shown in Fig. 2 and the known function of Ubch7 as a physiological E2 for RBRs, it is unclear why these results were included. If they were to remain in the paper, similar activity assays as those shown in Fig. 1F and 1G should be shown for other E2s.
2. In Fig. 2C, Cul2 was not neddylated. Given the importance of neddylation of Cul5 for interaction with TRIAD1, the lack of Cul2-neddylation makes it difficult to conclude that Triad1 specifically interacts with Cul5. The authors should discuss these findings more carefully.
3. Fig. 5E/F: it is difficult to draw conclusions about HHARI autoubiquitylation using SYPRO Ruby stains - the Cull1-bands also appear to be modified. For cleaner analysis, a Western blot detecting

HHARI should be shown.

Referee #3:

What is clear about the submitted work is that the authors have uncovered a very interesting phenomenon wherein RBR proteins of the Ariadne subfamily interact specifically with Nedd8-conjugated cullin-RING ubiquitin ligases. This interaction is not only conserved between two members of the Ariadne family (Triad1 and HHARI), but it appears to be relatively specific in that Triad1 only binds Cul5 whereas HHARI binds other CRLs. Unexpectedly, this interaction strongly favors the Nedd8-conjugated CRLs.

What is completely unclear about this manuscript is, what is the significance of the observed interactions? Unfortunately, this is not a straightforward question to address, because there are few if any well-characterized substrates available for Triad1 and Cul5, and hence the authors do not have an obvious and easy route to addressing functional significance. Thus, it is an editorial decision as to whether it is suitable for EMBO Journal. This reviewer is of the opinion that the work could potentially be suitable for publication in EMBO J given that it is a first description of an interesting and unexpected phenomenon that links together two important families of ubiquitin ligase enzymes. However, in their desire to highlight functional significance the authors have chased phenomena that are not compelling. The paper would be stronger if the authors refocused on a more detailed biochemical characterization of the interaction and its consequences for Triad1 activity.

Specific comments follow:

1. The authors argue that CRL5 activates Triad1. The problem with this is that the authors use two surrogate read-outs for activity, neither of which is known to be of relevance to function. In the presence of CRL5, the authors observe ubiquitination of Cul5 and enhanced autoubiquitination of Triad1. Ubiquitination of Cul5 - which is not particularly impressive - could be a trivial consequence of CRL5-Triad1 interaction. Enhanced autoubiquitination of Triad1 could result from a conformational change in Triad1 upon binding of CRL5, that favors transfer of ubiquitin to lysines within Triad1. This need not be due to enhanced ubiquitin ligase activity per se. Optimally one would like to see the effect of CRL5 on ubiquitination of an authentic Triad1 substrate. But since that seems unlikely to be feasible, an alternative would be to evaluate Triad1-mediated discharge of donor ubiquitin from UbcH7 or Ube2T to a large excess of acceptor ubiquitin to form diubiquitin. A further point about the conclusion that CRL5 'activates' Triad1 is that it seems to be at odds with experiments in Fig. 6, wherein it is shown that mutation of the RING2 domain strongly increases association with Cul5 and promotes accumulation of neddylated Cul5 in cells. Another way to phrase this finding is that a functional RING2 domain represses association of Triad1 with neddylated CRL5. In the absence of functional RING2, Triad1 should not be able to form a thioester intermediate in ubiquitin transfer, which may result in the accumulation of E2~ubiquitin thioesters on the RING1 site. However the latter is not likely to be important since a RING1 mutant that is unable to bind UbcH7 still interacts with Cul5. Thus, the simplest hypothesis is that formation of the thioester intermediate on RING2 represses association with Cul5, and thus the complexes they observe in co-IP experiments are likely to represent inactive Triad1. This proposal, at face value, is at odds with the authors' hypothesis that Cul5 is a physiological activator. It would be informative to see if RING2 suppresses Triad1-Cul5 interaction in vitro with purified proteins in a manner that depends on UbcH7, E1, and ATP.

2. The authors speculate that Triad1 may regulate CRL5 through interaction with the neddylated complex. In support of this idea, expression of the RING2 mutant of Triad1 stabilizes the Nedd8 conjugated state of Cul5 by inhibiting deneddylation. Arguably, this is a trivial consequence of the mutant protein binding neddylated Cul5 and shielding it from CSN. There is no evidence that this occurs normally, and in fact in cells missing Triad1 there is no effect on Cul5 neddylation status.

Detailed comments:

3. On page 4 where the authors cite Alexandru and den Besten they should also cite Bandau et al 2012.

4. page 6, bottom: there is no Figure 1J. Should read Figure 1I.

5. Page 15, first paragraph of Discussion: there is no evidence that, "RING2 domain is required to maintain a dynamic steady state level of the neddylated and active form of the CRLs".

6. Figure 3G: it would be nice to see a control that the beads have equivalent activity for binding a non-selective UBA or UIM domain. The authors should also show that their UBA mutant is defective in binding to these beads. As it stands the data in 3F aren't entirely convincing because the level of expression of the functional constructs may be considerably higher than for the non-functional constructs. Although this is not immediately apparent because the band on the western is too 'burned in', it is implied from looking at the bands that migrate immediately above and below the main band.

Additional correspondence (author)

22 March 2013

Please find attached a brief overview how we aim to address the referees' concerns and a response to each major points raised by the referees. I am happy to discuss any further details on the phone.

Thank you again for giving us the opportunity to respond to the referees' comments on our manuscript, "TRIAD1 and HHARI bind to and are activated by distinct neddylation Cullin RING ligase complexes".

All three Reviewers have few concerns about our key findings regarding the interaction of Ariadne RBRs with distinct neddylation CRL complexes. However, referees 2 and 3 are not fully convinced by the data on TRIAD1's stimulation of CUL5 activity and on the role of TRIAD1/HHARI in regulating Cullin neddylation and activity. To deal with these concerns we propose the following:

1.) We can improve the functional studies.

- To strengthen the functional relevance of the HHARI/CUL1 interaction, we can provide new data showing that HHARI enhances the ubiquitylation of cyclin E by CUL1-RBX1 *in vitro* and that cells expressing mutant HHARI-C357S stabilize cyclin E in cells.
- CUL5 has been implicated in the degradation of APOBEC3 as part of a HIV viral response. We are currently setting up assays to study APOBEC3 degradation to address whether TRIAD1/CUL5 functional interaction is required in this process.

2.) We can make use of alternative biochemical assays to better characterize cullin-stimulated RBR-ligase activity.

- Regarding cullin-stimulation of TRIAD1 and HHARI ligase activities, we will use an improved cullin ubiquitylation assay. This assay allows us to detect multiple ubiquitylation on the N terminal domain of the cullin. This assay will be used to study the time-course of ubiquitylation and to test different mutant versions of TRIAD1 and HHARI.

Here we provide a more detailed response to the **Major points** addressed by the referees:

Referee #1

(Remarks to the Author)

*The manuscript by Kellsall et al. aims to provide insight into the function and regulation of members of the RBR ligase family which have recently been shown to form the new family of RING/HECT hybrids that combine the activities of both, RING and HECT-type ligases. The authors make the interesting observation that the RBR ligases TRIAD1 and HHARI associate with neddylation cullin-RING ligase complexes and that this interaction positively modifies their ligase activity as judged by auto-ubiquitination assays. Furthermore, they map the region in the RBRs responsible for the interaction to a conserved acidic region and an UBA-like domain that specifically recognize NEDD8 and provide evidence that the RBR ligase activity in turn regulates CRL activity. **This is a very interesting story that identifies yet another level of regulation within the ubiquitination system and should be of great interest to a broad audience.***

Response: Referee 1 is overall very positive describing our work as “very interesting” and “of great interest to a broad audience”.

However, the authors should address a few issues to make their manuscript easier accessible and strengthen their model.

Major points

1- To identify cognate E2s for TRIAD1 the authors carried out IPs of endogenous TRIAD1. Not all E3s form stable complexes with their cognate E2s and hence the authors may have missed physiologically important E2s with this approach. It would be good if they added a comment explaining why they didn't try other approaches and if they think there might be other E2s that work with TRIAD1.

Response: The referee is quite right that not all E3s form stable complexes with E2s and part of the reason for showing this data must surely be to show how unusual TRIAD1 is in forming strong E2-E3 interactions such as with UBE2T and UBCH7. We cannot exclude having missed other physiological E2s and we will stress this point in the Discussion. However, we confirmed TRIAD1's interaction with UBE2T and UBCH7 using *in vitro* and *in vivo* binding and ubiquitylation assays. UBCH7 has recently been suggested to be the physiologic relevant E2 for RBR ligases (Wenzel DM, 2011). Moreover, though the data is not included, we have performed screens of all E2s to see whether they can work with TRIAD1 in auto-ubiquitylation assays and thus far only see significant activity with UBE2T and UBCH7. These data can be included in the revised manuscript.

More importantly, the authors carry out some of the experiments with UBE2T, others with UbCH7 and some with both E2s. Why not do everything with both?

Response: We focused our experiments with CRLs on TRIAD1/UBCH7 and HHARI/UBCH7 because UbCH7 was reported to transfer ubiquitin only to a cysteine acceptor as in HECT or RBR E3 enzymes, and does not work with CRLs. This allows us to observe ubiquitin ligation mediated by TRIAD1 or HHARI without complication of E3 ligase activity from CRLs.

*Did they try to detect the thioester using both E2s or only UbCH7? And do they think there's a functional difference between the two E2s? Please comment
In this respect, why didn't they find UBE2T in the IP MS experiment?*

Response: Referee 1 raised a valid point because we only used UBCH7 in assays aiming to detect Ub-thioester formation on TRIAD1. We can certainly repeat this assay using UBE2T. As for the IP MS experiment, UBE2T is a small protein and produces only a few peptides, which could well be swamped in a MS experiment. However, importantly, in a reciprocal IP MS experiment performed in the lab (though not shown in the paper) TRIAD1 was identified following UBE2T precipitation.

2- The authors show that TRIAD1 binds tighter to neddylated CUL5-RBX2 than the non-neddylated version and that TRIAD1 can even bind to isolated NEDD8. Given these observations it would be very interesting to see pull-downs directly comparing the relative affinities of TRIAD1 for NEDD8 versus neddylated CUL5-RBX2, i.e. does CUL5 strengthen the interaction or is the interface dominated by NEDD8?

Response: We show a comprehensive domain analysis of the TRIAD1/CUL5 and HHARI/CUL1 interaction *in vivo*. We concluded that 1) the interactions are strictly dependent on neddylated cullin 2) require the “basic canyon” on cullin C terminal domain 3) are strictly dependent on the acidic N terminus of TRIAD1 (HHARI) and 4) require the UBA-like domain in TRIAD1 (HHARI) and suggest cooperative binding sites. Given the many proteins involved, a functional dissection of these complexes using *in vitro* studies will be difficult. Direct pull down experiments as suggested by referee 1 are not feasible because TRIAD1 only binds to NEDD8 conjugated agarose beads.

And how does isolated NEDD8 affect auto-ubiquitination? Can it activate on its own?

Response: This is indeed a very interesting question. We already performed this experiment showing that NEDD8 alone does not affect HHARI auto-ubiquitylation, see Figure 5F.

3- The authors show convincingly that the interaction between TRIAD1 and neddylated CUL5-RBX2 stimulates the E3 ligase activity of TRIAD1. But what's the mechanism behind this? As the authors point out themselves in the Discussion this could be due to auto-inhibitory interactions as shown previously for PARKIN and HOIP. This should be tested in detail.

Response: Referee 1 clearly states that “the authors show convincingly” that neddylated cullin stimulates TRIAD1 E3 ligase activity. We are currently working on a crystal structure of auto-inhibited HHARI/TRIAD1 and trying to understand the structural basis for relief of auto-inhibition. However, we believe that a crystal structure of auto-inhibited HHARI or TRIAD1 is beyond the scope of this original manuscript. We hope that by the time we would submit a revised manuscript we could include structural information to the reviewers only that would explain the basis for auto-inhibition.

What is the difference, if any, of the auto-ubiquitination activity of TRIAD1 in the presence and absence of the region N-terminal to the RBR?

Response: This is a very important question and we indeed addressed it in Figure 4E. We show that removal of the N terminus (acidic plus UBA-like domains) abolish CUL5 ubiquitylation.

Are there differences between UbcH7 and UBE2T? And how does this compare to HHARI?

Response: All assays in Figure 4 have been carried out with UBCH7. We will repeat these assays with UBE2T to address referee’s question whether there is a difference between UBCH7 and UBE2T.

Referee #2

(Remarks to the Author)

*The RBR-family of E3s was recently shown to catalyze ubiquitylation via an obligate thioester intermediate, using a Cys residue in its RING2-domain. Although some members of the RBR-family, such as the E3s Parkin or LUBAC, are heavily studied, most RBRs remain rather poorly understood. **Determining mechanisms that underlie the function or regulation of RBRs would be important and interesting for a wide readership.***

In this manuscript, Kellsall et al. investigated the biochemical function of Triad1 and Hhari, two members of the Ariadne-class of RBRs. Biochemical analysis suggested that Triad1 acts as a canonical RBR, although a thioester-bond between a critical Cys residue in its RING2 and ubiquitin could not be detected. In cells, Triad1 was found to interact specifically with the neddylated form of Cul5, but not with other Cullin-RING ligases. HHARI similarly interacted with neddylated cullins, although it was less specific (with the exception that, interestingly, it did not interact with Cul5). The interactions between Triad1/HHARI and N8-Cul were direct and mediated via a UBA-domain in Triad1 and HHARI. Up to this part, this paper is very nice, and I have only very minor issues, as described below.

Response: We are pleased to see that referee 2 is convinced by our key finding showing conserved interactions of Ariadne RBR ligases with neddylated CRL complexes.

However, the authors next analyze the functional consequences of the interaction between Triad1 and N8-Cul5, and I found this part of the paper a bit unsatisfying. They report an increase in the apparent ubiquitylation activity of Triad1, if Cul5 was present. Unfortunately, although Triad1 appears to bind specifically to neddylated Cul5, the in vitro assays shown in Fig. 4a reveal only minor differences between unneddylated and neddylated Cul5, raising questions about the specificity of this assay.

Response: Referee 2 addresses a valid point, however, we can provide a reasonable explanation for this apparent contradiction. IP experiments and gel filtration experiments with recombinant purified proteins showing a strict requirement of NEDD8 for the TRIAD1/CUL5 interaction. These experiments rely on stable protein interactions. Our interpretation is that NEDD8 is indeed required

to stabilize the interaction, but can also occur in the absence NEDD8 (indeed, we observe a weak association between TRIAD1 and CUL5-RBX2 using *in vitro* pull down experiments see Fig. 3C). Importantly, Fig 4A shows a dramatic increase of TRIAD1 auto-ubiquitylation activity in the presence of CUL5-RBX2 and is further enhanced by neddylation of CUL5-RBX2. The interpretation of these data should be better seen in context of the *in vitro* binding data from Fig 3C.

One must also consider that such *in vitro* experiments can only ever be an approximation of the *in vivo* situation and we may be lacking other factors that further tailor the specificity of this interaction and the consequent stimulation of TRIAD1 activity only to the neddylation of the cullin.

Raising similar concerns, N8-Cul5 only allows addition of one extra ubiquitin to Triad1 in autoubiquitylation assays - in the absence of a known function of Triad1 autoubiquitylation, it is unclear whether this event is significant. Whether Cul5 really stimulates Triad1 activity requires either an analysis of Triad1-substrate ubiquitylation or, at least, a more in-depth biochemical analysis of the autoubiquitylation or chain formation (i.e. quantitative kinetics).

Response: We hope to mellow the concerns by pointing to Fig. 4C, which shows a clear formation of a ladder of mono-, di-, and tri-ubiquitin on TRIAD1 in a time dependent manner. We note that recent publications have used autoubiquitylation assays as a functional readout for RBR ligases such as HHARI and TRIAD1 without objection.

Though less significant, we observe ubiquitin conjugations on CUL5.

We have also exploited the “split CUL5” (separate expression and self assembly of the N- and C-terminal domain to form a fully functional CUL5) established by the Brenda Schulman lab (see CUL1-based assays used in Fig. 5). Preliminary data suggest that TRIAD1 ubiquitylates the N-terminal domain of CUL5 at multiple sites. We can further exploit this system for time-course assays.

In contrast to these rather weak effects on Triad1-autoubiquitylation, the authors show that Triad1 catalyzes monoubiquitylation of Cul5, and given their later findings, it is possible that this reaction plays a role in regulating the CRL. Is monoubiquitylation of Cul5 dependent on neddylation?

Response: *In vivo* the interaction between TRIAD1 and CUL5 is entirely dependent on neddylation and indeed here in Figure 4E, when we disrupt the interaction by cleaving off the N-terminus of TRIAD1, we see no modification of CUL5.

Utilizing the “split CUL5” we have preliminary data showing that ubiquitylation of CUL5 can occur in the absence of neddylation, although this was enhanced by addition of NEDD8. We can certainly follow up on these studies.

What is the site of this reaction (they have all components reconstituted, so should be able to determine the modification site by mass spectrometry)? Does interfering with this modification affect the activity of Cul5?

Response: This can be done.

Because very little is known about physiological substrates of Triad1 or HHARI, analyzing consequences of TRIAD1 on CRL-substrate ubiquitylation or degradation provides a more straightforward means to assessing the physiological importance of the reported interaction (which, given a recent Nature Immun. Paper on Triad1 is clear!). Therefore, experiments describing SCF- or Cul5-substrate modification and stability in the presence of HHARI or Triad1 could greatly strengthen the paper.

Response: As pointed out by referees 2 and 3, CUL5 substrates are not well-described, therefore we have focused our efforts on CUL1 substrates. In Fig. 6H we provide evidence that the CUL1 substrate p27^{Kip} is stabilized in HHARI C357S expressing cells. In addition we have already gained further data to address this point. We observe a stimulation in the ubiquitylation of cyclin E and p27^{Kip} by neddylation of CUL1 in the presence of HHARI *in vitro* (Figure A and B). We can further

support this data, by showing a modest, but significant, defect in degrading phospho-cyclin E in cells expressing HHARI C357S (Figure C).

Figure legend (A) Time course experiments of cyclin E peptide ubiquitylation by fully neddylated CUL1-RBX2 complex and CDC34 (N8-CUL1-RBX1/CDC34) in the absence (-) or presence (+) of HHARI. N8-CUL1-RBX1/CDC34 mediated ubiquitylation of cyclin E peptide detected by immunoblot analysis. (B) Time course experiment of N8-CUL1-RBX1/CDC34 mediated ubiquitylation of phospho-p27 in the absence (-) or presence (+) of HHARI. Phospho-p27 ubiquitylation [p27(Ub)_n] was detected by immunoblotting with anti p27 antisera. (C) The indicated cell lines were synchronized with 100 ng/ml nocodazole then released into fresh medium for the indicated times periods before western blotting with the indicated antibodies.

Referee #3

(Remarks to the Author)

What is clear about the submitted work is that the authors have uncovered a very interesting phenomenon wherein RBR proteins of the Ariadne subfamily interact specifically with Nedd8-conjugated cullin-RING ubiquitin ligases. This interaction is not only conserved between two members of the Ariadne family (Triad1 and HHARI), but it appears to be relatively specific in that Triad1 only binds Cul5 whereas HHARI binds other CRLs. Unexpectedly, this interaction strongly favors the Nedd8-conjugated CRLs.

Response: We are glad that the referee finds this “a very interesting phenomenon” that we have uncovered.

What is completely unclear about this manuscript is, what is the significance of the observed interactions? Unfortunately, this is not a straightforward question to address, because there are few if any well-characterized substrates available for Triad1 and Cul5, and hence the authors do not have an obvious and easy route to addressing functional significance.

*Thus, it is an editorial decision as to whether it is suitable for EMBO Journal. This reviewer is of the opinion that **the work could potentially be suitable for publication in EMBO J given that it is a first description of an interesting and unexpected phenomenon that links together two important families of ubiquitin ligase enzymes.***

However, in their desire to highlight functional significance the authors have chased phenomena that are not compelling. The paper would be stronger if the authors refocused on a more detailed biochemical characterization of the interaction and its consequences for Triad1 activity.

Specific comments follow:

1. The authors argue that CRL5 activates Triad1. The problem with this is that the authors use two surrogate read-outs for activity, neither of which is known to be of relevance to function.

Response: The referee is completely correct (and we certainly agree) to say that neither of our readouts is known to be of relevance to function. That said, we want to highlight that both readouts suggest an increase in TRIAD1's ligase activity in the presence of neddylated cullins.

In the presence of CRL5, the authors observe ubiquitination of Cul5 and enhanced autoubiquitination of Triad1. Ubiquitination of Cul5 - which is not particularly impressive - could be a trivial consequence of CRL5-Triad1 interaction.

Response: We agree, we cannot rule out this possibility. As mentioned in our response to referee 2, we recently used the "split CUL5" system, and observed a more pronounced ubiquitylation of the N terminal domain of CUL5. We can use this assay for a detail analysis of CUL5 ubiquitylation for the revision.

Figure legend: TRIAD1 ligase activity is assayed in the presence of CUL5-RBX2 and neddylated CUL5-RBX2 using "split CUL5". On SDS-PAGE CUL5 will separate as N- and C terminal domain. N terminal CUL5 ubiquitylation is analyzed using a CUL5 N terminal specific antibody.

Enhanced autoubiquitination of Triad1 could result from a conformational change in Triad1 upon binding of CRL5, that favors transfer of ubiquitin to lysines within Triad1. This need not be due to enhanced ubiquitin ligase activity per se.

Optimally one would like to see the effect of CRL5 on ubiquitination of an authentic Triad1 substrate. But since that seems unlikely to be feasible, an alternative would be to evaluate Triad1-mediated discharge of donor ubiquitin from UbCH7 or Ube2T to a large excess of acceptor ubiquitin to form diubiquitin.

Response: We tried very hard to set up TRIAD1 (and HHARI)-mediated discharge assays but without any success. Hence we are limited to auto-ubiquitylation (the standard assay for most RBRs characterized so far!) and cullin ubiquitylation assays. We also want to stress the point that these assays are used for both TRIAD1 and HHARI and that both ligase activities are stimulated by binding to their cognate cullins, suggesting a conserved phenomenon.

A further point about the conclusion that CRL5 'activates' Triad1 is that it seems to be at odds with experiments in Fig. 6, wherein it is shown that mutation of the RING2 domain strongly increases association with Cul5 and promotes accumulation of neddylated Cul5 in cells.

Another way to phrase this finding is that a functional RING2 domain represses association of Triad1 with neddylated CRL5. In the absence of functional RING2, Triad1 should not be able to form a thioester intermediate in ubiquitin transfer, which may result in the accumulation of E2~ubiquitin thioesters on the RING1 site. However the latter is not likely to be important since a RING1 mutant that is unable to bind Ubch7 still interacts with Cul5.

Thus, the simplest hypothesis is that formation of the thioester intermediate on RING2 represses association with Cul5, and thus the complexes they observe in co-IP experiments are likely to represent inactive Triad1. This proposal, at face value, is at odds with the authors' hypothesis that Cul5 is a physiological activator. It would be informative to see if RING2 suppresses Triad1-Cul5 interaction in vitro with purified proteins in a manner that depends on Ubch7, E1, and ATP.

Response: Here we would like to clarify our results shown in figure 6A and 6F. We do not necessarily agree that mutated TRIAD1-C310S and HHARI-C357S bind stronger to CUL5 and CUL1 respectively. Rather, importantly, cells expressing these RING2 mutants have much higher levels of neddylated cullins and hence, we observe more immunoprecipitated cullins.

2. The authors speculate that Triad1 may regulate CRL5 through interaction with the neddylated complex. In support of this idea, expression of the RING2 mutant of Triad1 stabilizes the Nedd8 conjugated state of Cul5 by inhibiting deneddylation. Arguably, this is a trivial consequence of the mutant protein binding neddylated Cul5 and shielding it from CSN. There is no evidence that this occurs normally, and in fact in cells missing Triad1 there is no effect on Cul5 neddylation status.

Response: We tested the idea whether HHARI protects neddylated cullin from deneddylation by CSN. Preliminary results suggest that even a 10x molar excess of HHARI did not prevent deneddylation of N8-CUL1 by CSN. We will include these data in the revised manuscript.

Summary: Taken together, we believe that we are in a strong position to address all issues raised by the reviewers. Through the use of ‘split-n-coexpress’ cullin proteins we are able to complement and improve upon the criticized biochemical studies and address the concerns expressed by the reviewers relating to the increase in TRIAD1 ligase activity. In addition, a combination of the *in vitro* substrate ubiquitylation assays and expanded *in vivo* analysis of cullin substrate stability should address the concerns expressed by reviewers 2 and 3 about the physiological importance of the observed RBR-cullin interaction. We already have such assays and techniques in place within the lab, and preliminary data included here support our current model of TRIAD1/HHARI as regulators of cullin-RING ligase activity.

1st Editorial Decision

22 March 2013

Thank you for response to the referee comments I forwarded you last week on your recent submission. I have now had a chance to consider your responses, and I am glad to see that you may be in a good position to address the key concerns with the functional analyses raised by all three referees. Combined with the overall interest of the findings, we shall therefore in principle be happy to consider a new version of this manuscript, incorporating the proposed revision experiments, further for publication. I should nevertheless stress that eventual acceptance of the paper will of course also depend on convincing the referees through your responses and additional data, and please also be reminded that it is our policy to allow a single round of major revision only, making it important to diligently and comprehensively answer to all the points raised at this stage in the process. When revising the manuscript, please also carefully proofread it to improve clarity, and adjust especially the reference formatting to our journal's format. Furthermore, please check for appropriate referencing and introduction of relevant recent publications and concepts, e.g. previous reports on thioester intermediates in other RBR ligases, or on TRIAD1-E2 interactions.

We generally allow three months as standard revision time, and it is our policy that competing manuscripts published during this period will have no negative impact on our final assessment of your revised study. However, we request that you contact us as soon as possible upon publication of any related work, to discuss how to proceed. Should you foresee a problem in meeting this three-month deadline, please let us know in advance and we may be able to grant an extension. I would also appreciate if you could keep me updated about the progress of the revision work and your tentative timeline for resubmission.

Thank you again for the opportunity to consider this work, and please do not hesitate to contact me in case you should have any additional question regarding this decision or the reports. I look forward to your revision.

Point-by-point response

We addressed in much detail the majority of the referees' points and we have completely rewritten the results section in the revised manuscript providing additional data that we feel strengthen our conclusions.

1.) In particular we aimed to incorporate and to build on the recent finding obtained from the structural analysis of human HHARI by the lab of Brenda Schulman (Duda, et al, 2013), which uncovered a mechanism of HHARI's auto-inhibition setting our discovery into a new perspective. Our work now proposes a mechanism whereby HHARI is relieved from its auto-inhibited configuration by binding to specific neddylated CRL complexes. Similar results were obtained in TRIAD1 activity studies suggesting a conserve regulatory mechanism for Ariadne RBR ligases. The discussion has also been rewritten to place our findings into context of the emerging theme, that RBR ligases, such as HOIP and PARKIN, are generally auto-inhibited and discuss the different but related modes of activation.

2.) To strengthen our conclusion that specifically the neddylated form of cullins stimulate Ariadne RBR ligase activities, we now provide additional biochemical assays assessing the ligase activity by different means. Besides auto-ubiquitylation assays we performed UBCH7~ubiquitin discharge assays (as suggested by referee 3) and reactivity with ubiquitin-vinyl-methyl ester (Ub-VME), the latter recently applied for assessing PARKIN activity (Wauer and Komander, 2013; Riley et al, 2013).

3.) Given the new focus that we placed on the biochemical analyses in the revised manuscript, we put less emphasis on the physiological study of Ariadne RBR/CRL interaction. However, we did improve the functional studies criticized by referees' 2 & 3. As suggested by referee 2 we focused our analyses on the HHARI/CUL1 interaction and extended CUL1 functional studies by analyzing the CUL1 substrate p27^{Kip1} as well as cyclin E. All together we feel that we strengthened our original observation and conclusion that HHARI influences the level of neddylated CUL1 and activity of CUL1 RING ligase complexes *in vivo*.

Here we provide the detailed response to the **major and minor points** addressed by the each referee:

Referee #1

(Remarks to the Author)

The manuscript by Kellsall et al. aims to provide insight into the function and regulation of members of the RBR ligase family which have recently been shown to form the new family of RING/HECT hybrids that combine the activities of both, RING and HECT-type ligases. The authors make the interesting observation that the RBR ligases TRIAD1 and HHARI associate with neddylated cullin-RING ligase complexes and that this interaction positively modifies their ligase activity as judged by auto-ubiquitination assays. Furthermore, they map the region in the RBRs responsible for the interaction to a conserved acidic region and an UBA-like domain that specifically recognize NEDD8 and provide evidence that the RBR ligase activity in turn regulates CRL activity.

This is a very interesting story that identifies yet another level of regulation within the ubiquitination system and should be of great interest to a broad audience. However, the authors should address a few issues to make their manuscript easier accessible and strengthen their model.

Major points

1- To identify cognate E2s for TRIAD1 the authors carried out IPs of endogenous TRIAD1. Not all E3s form stable complexes with their cognate E2s and hence the authors may have missed physiologically important E2s with this approach. It would be good if they added a comment explaining why they didn't try other approaches and if they think there might be other E2s that work with TRIAD1. More importantly, the authors carry out some of the experiments with UBE2T, others with UbcH7 and some with both E2s. Why not do everything with both? Did they try to detect the thioester using both E2s or only UbcH7? And do they think there's a functional difference between the two E2s? Please comment

In this respect, why didn't they find UBE2T in the IP MS experiment?

Response: In order to address the reviewer's concerns that we may have missed physiologically important E2s we undertook a screen of 34 different E2 conjugating enzymes using a TRIAD1 auto-ubiquitylation assay, the results of which are now included as **Supplementary Figure S1A**. We found that only UBCH7 displayed significant TRIAD1-dependent activity, in keeping with its proposed role as the cognate E2 for the whole RBR ligase family. In light of the issues raised by Reviewer 1, and the similar issues raised by Reviewer 2, we felt that the inclusion of data on UBE2T was a distraction from what we wished to be the central message of our data: the stimulation of TRIAD1/HHARI ligase activity upon binding to cullin RING ligase complexes. We have therefore taken the decision to remove the UBE2T data and have replaced it instead with a more in-depth biochemical analysis of TRIAD1's activity in the presence of UBCH7, highlighting TRIAD1's intrinsically low activity in the absence of CRL binding (see newly added **Figures 1D and 1E**). We hope that the changes are acceptable and improve the overall clarity of the manuscript.

2- The authors show that TRIAD1 binds tighter to neddylated CUL5-RBX2 than the non-neddylated version and that TRIAD1 can even bind to isolated NEDD8. Given these observations it would be very interesting to see pull-downs directly comparing the relative affinities of TRIAD1 for NEDD8 versus neddylated CUL5-RBX2, i.e. does CUL5 strengthen the interaction or is the interface dominated by NEDD8? And how does isolated NEDD8 affect auto-ubiquitination? Can it activate on its own?

Response: Direct pull down experiments, such as those described by the reviewer, are regrettably not feasible because TRIAD1 fails to bind to free NEDD8 and only binds to NEDD8 that has been conjugated to agarose beads (such as in the experiment shown in **Figure 3F**). Analytical sizing experiments (shown below) similarly fail to observe co-purification of TRIAD1 and free NEDD8, and therefore serve to highlight the preferential associations with neddylated cullins shown in **Figures 3C, D and 5D**. In response to the reviewer's question about whether NEDD8 can activate ligase activity alone, we find that isolated NEDD8 has no effect on HHARI auto-ubiquitylation, TRIAD1-catalyzed UBCH7 discharge or binding of Ub-VME activity probe to the catalytic cysteine of either HHARI or TRIAD1 (**Figures 6B, 4A, 6F and 6G respectively**). This data, together with our comprehensive domain analysis of the TRIAD1/CUL5 and HHARI/CUL1 interactions *in vivo* (where we show preferential interactions with specific cullins, the requirement for the acidic N-terminus of TRIAD1 (HHARI), the requirement for the "basic canyon" on the cullin C-terminal domain and the important role of the UBA-like domain within the Ariadne family member), suggest cooperative binding sites in which the binding interface is dominated neither by the cullin nor by NEDD8 but rather requires both to be present in order to allow binding and ligase activation. We have changed the text accordingly to highlight that NEDD8 alone does not affect TRIAD1 or HHARI activity.

Figure legend: Gel filtration chromatography analysis of TRIAD1 mixed with NEDD8. Eluted proteins detected by spectrophotometry (280 nm).

3- The authors show convincingly that the interaction between TRIAD1 and neddylated CUL5-RBX2 stimulates the E3 ligase activity of TRIAD1. But what's the mechanism behind this? As the authors point out themselves in the Discussion this could be due to auto-inhibitory interactions as shown previously for PARKIN and HOIP. This should be tested in detail. What is the difference, if any, of the auto-ubiquitination activity of TRIAD1 in the presence and absence of the region N-terminal to the RBR? Are there differences between UbcH7 and UBE2T? And how does this compare to HHARI?

Response: These questions are best addressed in light of the structure of auto-inhibited HHARI recently solved by our collaborators in the Schulman laboratory (Duda *et al.* 2013 Structure of HHARI, a RING-IBR-RING ubiquitin ligase: auto-inhibition of an Ariadne-family E3 and insights into ligation mechanism. *Structure* **21**, 1030-1041). This work shows that HHARI, in the absence of a CRL binding partner, exists in an auto-inhibited state in which the C-terminal “Ariadne domain” embraces the surface of RING2 and occludes the catalytic Cys357. This auto-inhibition could be relieved by removing, or introducing point mutations into, this C-terminal domain of HHARI, and could be restored by reintroducing the isolated Ariadne domain. We are currently pursuing structural studies to better understand how cullin binding might influence Ariadne domain conformation, but this is currently at a very early stage. However, we provide new data in the revised manuscript (**Figure 6**), showing that neddylated CUL1 stimulates HHARI activity to an extent observed for a version of HHARI that lacks the auto-inhibitory Ariadne domain. We also demonstrate that those Ariadne domain mutants reported to relieve HHARI auto-inhibition *in vitro* dramatically alter HHARI protein stability *in vivo* and do so in a manner that is dependent on both HHARI catalytic activity and the presence of a functional proteosomal machinery. We have modified the text of our manuscript to incorporate a discussion of this very recent HHARI structural data where we propose a mechanism by which TRIAD1/HHARI auto-inhibition is relieved by binding to neddylated CRLs.

.... What is the difference, if any, of the auto-ubiquitination activity of TRIAD1 in the presence and absence of the region N-terminal to the RBR?

Response: This is a very important question and we one which we address in **Figure 4C**. We show that deletion of the N terminus (acidic domain and UBA-like domain) of TRIAD1 abolishes the CUL5-N8-induced activity of TRIAD1 to discharge ubiquitin from UBCH7~Ub.

Minor points

- Introduction page 4, it would be helpful for the reader if a sentence describing the function of mammalian HHARI was added.

Response: We agree that this would be helpful and have added the requested description of HHARI's reported functions in mammals.

- *Introduction: "...it remains to be addressed, how Ariadne RBR ligases can function as E3 ligases." What are the authors trying to say with this sentence? Are they referring to regulation of their activity? - Please be more specific.*

Response: This sentence was indeed poorly worded and has now been changed to... "Despite some evidence for the biological importance, the mechanisms regulating Ariadne RBR ligase function still remain poorly understood".

- *Introduction: "...distinct but specifically neddylated..." What do the authors mean? Could it be anything else than "specifically" neddylated?*

Response: The phrase "but specifically" has been removed.

- *On page 8 the authors mention the "basic canyon" and that more of the Elongin-B/-C adaptor complex co-precipitated with mutants that didn't bind to TRIAD1 anymore. Could you please add one or two explanatory sentences about the basic canyon (maybe a figure in Supplementary) and what the significance of the Elongin-B/-C adaptor complex co-IP is?*

Response: We have added the following description of the CUL5 basic canyon: "This positively-charged groove on the convex side of the cullin has been reported to bind to the acidic tail of the E2 enzyme Cdc34, although patterns of conservation suggest that multiple CRL cofactors may engage this particular surface feature". We have also added the following description of the Elongin-B/C complex... "We further noted that more of the neddylated form of CUL5 as well as more of the Elongin-B and Elongin-C substrate adaptor proteins, the essential subunits required for the binding of substrates to CUL5-based CRL complexes, co-precipitated with these mutant CUL5 versions".

- *It would be helpful to re-blot the auto-ubiquitination assay blots with the anti-TRIAD1 antibody.*

Response: This has been performed and is now incorporated into **Figure 4F** and **4G**.

- *This might seem very picky, but I'm not convinced that @ is a good symbol for a figure.*

Response: Agreed. This symbol has now been replaced by an asterisk *.

- *The alignment in Fig 3E would be much easier to understand if the authors used colour to highlight conserved residues.*

Response: This figure has been moved to **Supplementary Figure S3** and has been significantly increased in size. We hope that the conserved residues are better visible and the alignment understandable.

Referee #2

(Remarks to the Author):

The RBR-family of E3s was recently shown to catalyze ubiquitylation via an obligate thioester intermediate, using a Cys residue in its RING2-domain. Although some members of the RBR-family, such as the E3s Parkin or LUBAC, are heavily studied, most RBRs remain rather poorly understood. Determining mechanisms that underlie the function or regulation of RBRs would be

important and interesting for a wide readership.

In this manuscript, Kellsall et al. investigated the biochemical function of Triad1 and Hhari, two members of the Ariadne-class of RBRs. Biochemical analysis suggested that Triad1 acts as a canonical RBR, although a thioester-bond between a critical Cys residue in its RING2 and ubiquitin could not be detected. In cells, Triad1 was found to interact specifically with the neddylated form of Cul5, but not with other Cullin-RING ligases. HHARI similarly interacted with neddylated cullins, although it was less specific (with the exception that, interestingly, it did not interact with Cul5). The interactions between Triad1/HHARI and N8-Cul were direct and mediated via a UBA-domain in Triad1 and HHARI. Up to this part, this paper is very nice, and I have only very minor issues, as described below.

However, the authors next analyze the functional consequences of the interaction between Triad1 and N8-Cul5, and I found this part of the paper a bit unsatisfying. They report an increase in the apparent ubiquitylation activity of Triad1, if Cul5 was present. Unfortunately, although Triad1 appears to bind specifically to neddylated Cul5, the in vitro assays shown in Fig. 4a reveal only minor differences between unneddylated and neddylated Cul5, raising questions about the specificity of this assay. Raising similar concerns, N8-Cul5 only allows addition of one extra ubiquitin to Triad1 in autoubiquitylation assays - in the absence of a known function of Triad1 autoubiquitylation, it is unclear whether this event is significant. Whether Cul5 really stimulates Triad1 activity requires either an analysis of Triad1-substrate ubiquitylation or, at least, a more in-depth biochemical analysis of the autoubiquitylation or chain formation (i.e. quantitative kinetics).

Response: In light of the reviewer's concerns about the auto-ubiquitylation assays we use to characterize TRIAD1/HHARI's activation by CRLs, and the similar reservations expressed by Reviewer 3, we have now performed a more in-depth biochemical analysis, as requested. We have taken advantage of TRIAD1's ability to "discharge" ubiquitin from ubiquitin-loaded UBCH7 onto a large excess of acceptor lysine, and thank Reviewer 3 for suggesting such an experiment as an additional read-out of TRIAD1 activity. We have integrated such experiments into **Figure 1 and 4 (Figure 1D and 1E; Figure 4A, B, and D)** and believe that this type of assay offers extremely convincing data regarding TRIAD1's increased E3 ligase activity in the presence of neddylated CUL5. This assay provides a much more compelling result regarding the effects of neddylated versus non-neddylated cullin on TRIAD1's activity than the auto-ubiquitylation assay, with only neddylated cullin promoting TRIAD1-mediated ubiquitin discharge (**Figures 4A-C**).

Several attempts to assay NEDD8-CUL1-RBX1-induced HHARI activity using an UBCH7~ubiquitin discharge assay, as described for TRIAD1, failed. In search for an alternative method to assess the catalytic activity of HHARI in a more direct manner we made use of the recent discovery that the catalytic triad of RING2s resembles that of Cys-based deubiquitylases (DUBs) (Riley et al, 2013; Wauer & Komander, 2013). Based on this similarity with DUBs, the catalytic cysteine within RING2 can be modified by DUB suicide probe Ubiquitin vinylmethyl ester (Ub-VME). Using this assay we can show that NEDD8-CUL1-RBX1 greatly stimulates the formation of covalently Ub-VME-modified HHARI whereas CUL1-RBX1 has only a minor effect and free NEDD8 has no stimulatory effect (**Figure 6F**). Analogous experiments revealed that NEDD8-CUL5-RBX2 stimulated the reaction between TRIAD1 and Ub-VME (**Figure 6G**).

In summary, we use three different readouts – auto-ubiquitylation, UBCH7~ubiquitin discharge and Ubiquitin-vinylmethyl ester modification – to assess RBR ligase activities. All three assays suggest that neddylated cullins stimulate RBR ligase activity.

In contrast to these rather weak effects on Triad1-autoubiquitylation, the authors show that Triad1 catalyzes monoubiquitylation of Cul5, and given their later findings, it is possible that this reaction plays a role in regulating the CRL. Is monoubiquitylation of Cul5 dependent on neddylation? What is the site of this reaction (they have all components reconstituted, so should be able to determine

the modification site by mass spectrometry)? Does interfering with this modification affect the activity of Cul5? Because very little is known about physiological substrates of Triad1 or HHARI, analyzing consequences of TRIAD1 on CRL-substrate ubiquitylation or degradation provides a more straightforward means to assessing the physiological importance of the reported interaction (which, given a recent Nature Immun. Paper on Triad1 is clear!). Therefore, experiments describing SCF- or Cul5-substrate modification and stability in the presence of HHARI or Triad1 could greatly strengthen the paper.

Response: Given our increased focus on ubiquitin-discharge assays as a readout of TRIAD1 activity, and because of our agreement with Reviewer 3 that the significance and physiological relevance of the CUL5 monoubiquitylation is unclear, we have chosen to place less emphasis on those results, while still using them to complement the discharge, Ub-VME and auto-ubiquitylation results as a general indicator of TRIAD1 activity. Unfortunately mapping the *in vitro* modification site(s) on CUL5 by mass spec was not as straight-forward as might be envisaged, with no one site standing out as particularly significant, and with little overlap with those sites that we identified as being ubiquitylated in cells. We have therefore taken the reviewer's advice and analyzed the consequences of HHARI on CUL1 substrate ubiquitylation/stability as a means of assessing the physiological importance of the interaction. In addition to the *in vivo* data on endogenous p27^{Kip1} stability originally provided, we have now added data on the *in vivo* stability of another CUL1 substrate, Cyclin E. (**Figures 7F-H**). In the revised version we also provide data from cell cycle studies showing an accumulation of cyclin E in G1/S and an enrichment of p27^{Kip1} in M/G1 cell cycle phases of GFP-HHARI(C357S) expressing cells (**Figure 7G**).

Minor issues:

1. *Fig. 1B: different exposures between E2s make it almost impossible to judge the specificity of these IPs. Given the unbiased mass spectrometry shown in Fig. 2 and the known function of Ubch7 as a physiological E2 for RBRs, it is unclear why these results were included. If they were to remain in the paper, similar activity assays as those shown in Fig. 1F and 1G should be shown for other E2s.*

Response: As described earlier, we have elected to remove the data on E2s except UBCH7, which is, as the reviewer points out, commonly accepted as the physiological E2 for all RBR ligases. We do, however, include activity assays for over 34 E2s – see response to reviewer 1 - as a supplementary figure (**Supplementary Figure S1A**).

2. *In Fig. 2C, Cul2 was not neddylation. Given the importance of neddylation of Cul5 for interaction with TRIAD1, the lack of Cul2-neddylation makes it difficult to conclude that Triad1 specifically interacts with Cul5. The authors should discuss these findings more carefully.*

Response: We thank the reviewer for pointing this out. This experiment has been repeated and the new data, clearly showing CUL2 neddylation, is included in **Figure 2C**.

3. *Fig. 5E/F: it is difficult to draw conclusions about HHARI autoubiquitylation using SYPRO Ruby stains - the Cull1-bands also appear to be modified. For cleaner analysis, a Western blot detecting HHARI should be shown.*

Response: We have now blotted these assays with an antibody against HHARI and provide this data in addition to the original figures to allow clearer analyses (in the revised version **Figure 6C**).

Referee #3

(Remarks to the Author):

What is clear about the submitted work is that the authors have uncovered a very interesting phenomenon wherein RBR proteins of the Ariadne subfamily interact specifically with Nedd8-conjugated cullin-RING ubiquitin ligases. This interaction is not only conserved between two members of the Ariadne family (Triad1 and HHARI), but it appears to be relatively specific in that Triad1 only binds Cul5 whereas HHARI binds other CRLs. Unexpectedly, this interaction strongly favors the Nedd8-conjugated CRLs.

What is completely unclear about this manuscript is, what is the significance of the observed interactions? Unfortunately, this is not a straightforward question to address, because there are few if any well-characterized substrates available for Triad1 and Cul5, and hence the authors do not have an obvious and easy route to addressing functional significance. Thus, it is an editorial decision as to whether it is suitable for EMBO Journal. This reviewer is of the opinion that the work could potentially be suitable for publication in EMBO J given that it is a first description of an interesting and unexpected phenomenon that links together two important families of ubiquitin ligase enzymes. However, in their desire to highlight functional significance the authors have chased phenomena that are not compelling. The paper would be stronger if the authors refocused on a more detailed biochemical characterization of the interaction and its consequences for Triad1 activity.

Response: We thank the reviewer for their positive comments. As suggested, we have indeed attempted to refocus the paper on a more detailed biochemical characterization of TRIAD1/HHARI activity and strongly believe that this has significantly strengthened our work.

Specific comments follow:

1. The authors argue that CRL5 activates Triad1. The problem with this is that the authors use two surrogate read-outs for activity, neither of which is known to be of relevance to function. In the presence of CRL5, the authors observe ubiquitination of Cul5 and enhanced autoubiquitination of Triad1. Ubiquitination of Cul5 - which is not particularly impressive - could be a trivial consequence of CRL5-Triad1 interaction. Enhanced autoubiquitination of Triad1 could result from a conformational change in Triad1 upon binding of CRL5, that favors transfer of ubiquitin to lysines within Triad1. This need not be due to enhanced ubiquitin ligase activity per se. Optimally one would like to see the effect of CRL5 on ubiquitination of an authentic Triad1 substrate. But since that seems unlikely to be feasible, an alternative would be to evaluate Triad1-mediated discharge of donor ubiquitin from UbcH7 or Ube2T to a large excess of acceptor ubiquitin to form diubiquitin.

Response: We acknowledge that neither of the readouts we have used (CUL5 ubiquitylation or TRIAD1 auto-ubiquitylation) are likely to be of relevance to TRIAD1's physiological function and we thank the reviewer for suggesting ubiquitin discharge from UBCH7 as an alternative measure of TRIAD1 activity. As detailed earlier in the response to reviewer 2, such assays have now been incorporated into several figures (**Figure 1D and 1E; Figure 4A, B, and D**) and should assuage any doubts the reviewer has about the ability of neddylated CUL5 to stimulate TRIAD1 activity. These results are further complemented by assays using Ub-VME (Figure 6E-G).

A further point about the conclusion that CRL5 'activates' Triad1 is that it seems to be at odds with experiments in Fig. 6, wherein it is shown that mutation of the RING2 domain strongly increases association with Cul5 and promotes accumulation of neddylated Cul5 in cells. Another way to phrase this finding is that a functional RING2 domain represses association of Triad1 with neddylated CRL5. In the absence of functional RING2, Triad1 should not be able to form a thioester intermediate in ubiquitin transfer, which may result in the accumulation of E2~ubiquitin thioesters on the RING1 site. However the latter is not likely to be important since a RING1 mutant that is unable to bind UbcH7 still interacts with Cul5. Thus, the simplest hypothesis is that formation of the

thioester intermediate on RING2 represses association with Cul5, and thus the complexes they observe in co-IP experiments are likely to represent inactive Triad1. This proposal, at face value, is at odds with the authors' hypothesis that Cul5 is a physiological activator. It would be informative to see if RING2 suppresses Triad1-Cul5 interaction in vitro with purified proteins in a manner that depends on UbcH7, E1, and ATP.

2. The authors speculate that Triad1 may regulate CRL5 through interaction with the neddylated complex. In support of this idea, expression of the RING2 mutant of Triad1 stabilizes the Nedd8 conjugated state of Cul5 by inhibiting deneddylation. Arguably, this is a trivial consequence of the mutant protein binding neddylated Cul5 and shielding it from CSN. There is no evidence that this occurs normally, and in fact in cells missing Triad1 there is no effect on Cul5 neddylation status.

Response: To be more concise and focused we removed the data regarding CSN. However we performed experiments addressing the referee's comment and have included them below. As can be seen in the Figure, binding of HHARI (in 3-10x molar excess) to CUL1-NEDD8 offers no significant protection from de-neddylation *in vitro*.

Figure: Neddylated CUL1^{CTD} was de-neddylated in a reaction with CSN in the absence (mock) or the presence of 3x and 10x molar excess of HHARI. Reaction products were resolved on SDS-PAGE and subjected for immunoblot analysis with anti CUL1 antibody.

Detailed comments:

3. On page 4 where the authors cite Alexandru and den Besten they should also cite Bandau et al 2012.

Response: Quite correct. This has been corrected.

4. page 6, bottom: there is no Figure 1J. Should read Figure 1I.

Response: As figure 1 has been reworked this comment no longer applies.

5. Page 15, first paragraph of Discussion: there is no evidence that, "RING2 domain is required to maintain a dynamic steady state level of the neddylated and active form of the CRLs".

Response: This sentence has been re-phrased.

6. Figure 3G: it would be nice to see a control that the beads have equivalent activity for binding a non-selective UBA or UIM domain. The authors should also show that their UBA mutant is defective in binding to these beads. As it stands the data in 3F aren't entirely convincing because the level of expression of the functional constructs may be considerably higher than for the non-functional constructs. Although this is not immediately apparent because the band on the western is too

'burned in', it is implied from looking at the bands that migrate immediately above and below the main band.

Response: We have added binding data for the non-selective UIM domain-containing protein UBXN7 (**Figure 3F**). Unfortunately we were not able to express the UBA mutant protein in bacteria and attempts to purify the isolated domain also failed. It's true that there are differences in the TRIAD1 expression levels in Figure 3F, but we would argue that they are not as great as the differences seen in CUL5 binding. We would also point the reviewer in the direction of **Figure 5C**; this figure shows similar mutational analysis for HHARI and also indicates a loss of cullin binding upon disruption of the domain (this data can also now be viewed in light of the knowledge imparted by the HHARI crystal structure, which highlights the importance of the V123 residue in HHARI).

2nd Editorial Decision

26 August 2013

Thank you for submitting your revised manuscript and RBR/CRL functional interactions, and please excuse the delay associated with its re-evaluation during the summer vacation period. The three original referees have now once more looked into the study. As you will see from the enclosed reports, they find the biochemical and mechanistic analyses considerably improved, and potentially of general interest. They nevertheless retain concerns regarding the unclear physiological significance of the characterized interactions, as well as some specific presentational and control issues. Given their positive comments on the importance of the biochemical and mechanistic insights, we shall be happy to accept the manuscript pending answering/addressing these various specific comments, without requiring additional physiological analyses that may be best left for dedicated follow-up studies. Therefore, please revise the manuscript once more in response to the remaining specific points, including some discussions of the further-reaching points as well as a point-by-point response letter. Please also make sure to update the newly added citations, some of which are still incomplete. Following this final round of minor revision, we should hopefully be able to swiftly proceed with publication of this manuscript in our journal. Please do not hesitate to contact me should you have any questions in this regard. I look forward to receiving your final version!

REFEREE REPORTS:

Referee #1 :

The authors have made major changes to the manuscript in order to address all the issues raised by the reviewers. Their decision to put more emphasis on the biochemical analysis and to use the recently published structure of autoinhibited HHARI to interpret their data has significantly improved the manuscript. Therefore, I think that the data presented in this manuscript that provide the first link between RBR ligases of the Ariadne subfamily and neddylated cullins are now suitable for publication in EMBO J.

Nevertheless I have some remaining issues which I strongly urge the authors to address:

As shown in the recent publication by Duda et al. removal of the Ariadne domain releases autoinhibition and increases autoubiquitination of HHARI, in agreement with data presented in the current manuscript that removal of the Ariadne domain from HHARI allows modification with Ub-vinylmethyl ester.

Why did the authors not make the equivalent construct of TRIAD1 and tested for autoubiquitination as well as modification with Ub-VME? That seems an obvious control to support the conclusion that TRIAD1 is autoinhibited. Furthermore, such an "active" construct may also allow detection of a thioester.

The authors should add a sentence explaining as to why UBCH7~discharge assays could not be carried out with HHARI, otherwise it is confusing for the reader that a different assay was used for

HHARI than previously for TRIAD1.

On page 19 bottom the authors write that "...neddylated cullins relieves the Ariadne domain-mediated auto-inhibition, ..." That doesn't make any sense as the authors have shown that the N-terminal portion of TRIAD1/HHARI containing the acidic and UBA-like domain is responsible for the interaction with the neddylated cullin, not the C-terminally located Ariadne domain. This should be rephrased to fit with the data shown. For the same reason the portion of Figure 6 D depicting the cullin interaction doesn't make any sense - and in fact contradicts Figure 8 (which is in line with the data presented).

Furthermore, it would be beneficial to the reader if the authors added a cartoon of the domain structure of HHARI to Fig 6D; this is a complicated structure that is not very easy to understand.

Please correct the labelling of lanes in Figures 4A and 6A, B. They are shifted to the left.

Referee #2 :

The authors have addressed some of the issues that were raised in the first round of review. Although the biochemical part of this manuscript was already at a high level, the authors have further improved it, thus deepening our understanding of biochemically relevant CRL/RBR interactions. On the other hand, the paper still suffers from a lack of experiments that support a physiological role for this interaction (most experiments shown in the paper are overexpression studies). Even though the latter issue is somewhat disappointing, I believe that uncovering a biochemically relevant interaction between CRLs and RBRs will stimulate much work in this area, and given the high caliber of the *in vitro* analyses, I support publication in EMBO J.

Referee #3 :

In this revised manuscript, the authors clarify many of the mechanistic questions by carrying out a series of detailed biochemical assays and cell-based assays. The results are very interesting and imply a novel mode of E3 activation, which provides an elegant mechanistic insight for the organized activation of multiple ligases.

Specific comments follow:

1. The data shows clearly a small fraction of Triad1 binds specifically to the neddylated form of Cul5 and that HHARI binds other neddylated Cullins. The authors build on their previous experiments and more clearly show the effect of neddylated Cullins in activating Ariadne family proteins. The new experiments in Fig 4a and 4b, with UbcH7 show that neddylated-Cul5 helps to discharge UbcH7 in a Triad1 dependent manner that is not seen with free Nedd8 or unmodified Cul5. In fact, unmodified Cul5 and free Nedd8 appear to have no activity in these assays. What remains unclear about these results is that Fig 4D and 4E show that unmodified Cul5 is less than two-fold less effective in stimulating this activity. Although this is a bit murky, the cell-based IP assays seem to imply that binding to unmodified cullins is less significant *in vivo*.
2. The authors also do not rule out the concern that HHRAI and Triad1 can compete with CSN. If HHRAI and Triad1 binding is mediated by Nedd8, then the recent structural work by Enchev et al 2012 implies that they might have competitive steric interference with CSN. The deneddylation assays show that by 4 min the mock reaction is greater than 50% complete. Neither of the HHRAI reactions reaches this milestone within the 10 min reactions. This implies that HHRAI reduces neddylation rates by at least two-fold. This data is consistent with what was recently shown by Emberley et al 2012 to be the effect of substrate on inhibiting cullin deneddylation.
3. Although the biochemical work showing the increased activity of Triad1 and HHRAI upon binding is convincing, the biological significance is still unclear. Most of the striking effects are only seen by catalytically inactive mutants. For example, the deneddylation assay in Fig 7C, and the neddylation assay in Supplemental Fig 5D, show no effect of wildtype proteins. Even the effect on Cul1 substrates are only striking for the C357S mutant of HHRAI. Reading the text, it is not clear how the activation of one E3 Ligase (HHRAI) by a second E3 (Cul1) leads to the accumulation of the Cul1's substrates. There appears to be no evidence for increased ubiquitylation or degradation of

Cul1 or its substrates as by HHRAI. Also, HHRAI and Triad1 do not appear to be substrates of Cul1 or Cul5. It is not intuitive how activation of Ariadne proteins by cullins leads to regulation of cullin substrates based on the presented data. In the absence of Ariadne substrates, it is impossible to know the effect of cullins beyond that fact that they clearly bind and activate Ariadne subfamily proteins *in vitro*.

Detailed comment:

Supplemental Fig 5E. The cell synchronization data are not very convincing to the eye for G1 arrest. The authors should include percentages on the Flow graphs.

2nd Revision - authors' response

03 September 2013

Point-by-point response:

Referee #1 :

The authors have made major changes to the manuscript in order to address all the issues raised by the reviewers. Their decision to put more emphasis on the biochemical analysis and to use the recently published structure of autoinhibited HHARI to interpret their data has significantly improved the manuscript. Therefore, I think that the data presented in this manuscript that provide the first link between RBR ligases of the Ariadne subfamily and neddylated cullins are now suitable for publication in EMBO J.

Nevertheless I have some remaining issues, which I strongly urge the authors to address:

As shown in the recent publication by Duda et al. removal of the Ariadne domain releases autoinhibition and increases autoubiquitination of HHARI, in agreement with data presented in the current manuscript that removal of the Ariadne domain from HHARI allows modification with Ub-vinylmethyl ester.

Why did the authors not make the equivalent construct of TRIAD1 and tested for autoubiquitination as well as modification with Ub-VME? That seems an obvious control to support the conclusion that TRIAD1 is autoinhibited. Furthermore, such an "active" construct may also allow detection of a thioester.

Response: We totally agree with referee 1 that testing a construct of TRIAD1, which has the Ariadne domain removed, would be a good experiment. However, over the course of the past 6-8 months we have tried extensively to express a TRIAD1 deletion mutant without the Ariadne domain – including several different expression vectors, fusions even to solubility-enhancing proteins, etc, but unfortunately, we have not been able to make this protein. We have added a sentence to the revised manuscript explaining that inability to produce TRIAD1 lacking the Ariadne domain has prevented corresponding experiments with TRIAD1.

The authors should add a sentence explaining as to why UBCH7~discharge assays could not be carried out with HHARI, otherwise it is confusing for the reader that a different assay was used for HHARI than previously for TRIAD1.

Response: This is certainly a good experiment. Unfortunately, also despite extensive attempts, we only see HHARI auto-ubiquitylation in pulse-chase when we add UbCH7~Ub and neddylated cullins, even in the presence of excess lysine. We have added a sentence to the revised manuscript stating that rapid HHARI auto-ubiquitylation even in the presence of excess lysine prevented parallel experiments with HHARI.

On page 19 bottom the authors write that "...neddylated cullins relieves the Ariadne domain-mediated auto-inhibition, ..." That doesn't make any sense as the authors have shown that the N-terminal portion of TRIAD1/HHARI containing the acidic and UBA-like domain is responsible for the interaction with the neddylated cullin, not the C-terminally located Ariadne domain. This should be rephrased to fit with the data shown. For the same reason the portion of Figure 6 D depicting the

cullin interaction doesn't make any sense - and in fact contradicts Figure 8 (which is in line with the data presented).

Response: We apologize for the confusion, however, we feel that the sentence "...neddylated cullins relieves the Ariadne domain-mediated auto-inhibition, ..." is perfectly right and does explain our presented data. The referee might have misunderstood the message, perhaps as a result of the inaccurate placement of the cullin/RBX1 close to the Ariadne domain in the model presented in figure 6D. We acknowledge that this is indeed confusing and have decided to remove the cartoon from figure 6d, preferring instead to summarize our findings in figure 8. We have also rephrased sentences in the text to better explain and highlight the features of our model in figure 8.

Furthermore, it would be beneficial to the reader if the authors added a cartoon of the domain structure of HHARI to Fig 6D; this is a complicated structure that is not very easy to understand.

Response: Indeed the structure of HHARI may be difficult for a non-structural biologist and we thank the referee for the suggestion. However, as stated, we have removed the cartoon for reasons mentioned above.

Please correct the labelling of lanes in Figures 4A and 6A, B. They are shifted to the left.

Response: The labeling has been corrected in Figures 4A, 6A and 6B.

Referee #2 :

The authors have addressed some of the issues that were raised in the first round of review. Although the biochemical part of this manuscript was already at a high level, the authors have further improved it, thus deepening our understanding of biochemically relevant CRL/RBR interactions. On the other hand, the paper still suffers from a lack of experiments that support a physiological role for this interaction (most experiments shown in the paper are overexpression studies). Even though the latter issue is somewhat disappointing, I believe that uncovering a biochemically relevant interaction between CRLs and RBRs will stimulate much work in this area, and given the high caliber of the in vitro analyses, I support publication in EMBO J.

Response: We are pleased that referee 2 recognizes the improvement of our biochemical and mechanistic analyses and that they meet the high quality standard/scientific impact to make it suitable for publication in EMBO J.

Referee #3 :

In this revised manuscript, the authors clarify many of the mechanistic questions by carrying out a series of detailed biochemical assays and cell-based assays. The results are very interesting and imply a novel mode of E3 activation, which provides an elegant mechanistic insight for the organized activation of multiple ligases.

Specific comments follow:

1. The data shows clearly a small fraction of Triad1 binds specifically to the neddylated form of Cul5 and that HHARI binds other neddylated Cullins. The authors build on their previous experiments and more clearly show the effect of neddylated Cullins in activating Ariadne family proteins. The new experiments in Fig 4a and 4b, with UbcH7 show that neddylated-Cul5 helps to discharge UbcH7 in a Triad1 dependent manner that is not seen with free Nedd8 or unmodified Cul5. In fact, unmodified Cul5 and free Nedd8 appear to have no activity in these assays. What remains unclear about these results is that Fig 4D and 4E show that unmodified Cul5 is less than two-fold less effective in stimulating this activity. Although this is a bit murky, the cell-based IP assays seem to imply that binding to unmodified cullins is less significant in vivo.

Response: We applied three different assays to assess TRIAD1 and HHARI activity. Importantly, all these assays agree that neddylated cullins stimulate TRIAD1 and HHARI activity. These results

are also supported by data showing that TRIAD1 and HHARI interact exclusively (within the given sensitivity of the IP experiment) with the neddylated cullin *in vivo*. The modest stimulation of TRIAD1 and HHARI by non-neddylated cullin is only observed by the *in vitro* auto-ubiquitylation assay, which we regard as a more sensitive assay in comparison to the UBCH7~ubiquitin discharge assay. As we do not observe any detectable interaction of TRIAD1 or HHARI with non-neddylated cullins *in vivo*, we consider the stimulation by non-neddylated cullins likely to be less significant *in vivo* and in fact limited to *in vitro* conditions.

2. *The authors also do not rule out the concern that HHRAI and Triad1 can compete with CSN. If HHRAI and Triad1 binding is mediated by Nedd8, then the recent structural work by Enchev et al 2012 implies that they might have competitive steric interference with CSN. The deneddylation assays show that by 4 min the mock reaction is greater than 50% complete. Neither of the HHRAI reactions reaches this milestone within the 10 min reactions. This implies that HHRAI reduces neddylation rates by at least two-fold. This data is consistent with what was recently shown by Emberley et al 2012 to be the effect of substrate on inhibiting cullin deneddylation.*

Response: At this point, we have not been able to convince ourselves of a reproducible, substantial effect on CSN-mediated *in vitro* de-neddylation. We see similar effects of adding HHARI and TRIAD1 to the reactions, raising the possibility that differences are artefacts from the extra protein in the assays. The effects of HHARI and TRIAD1 on de-neddylation of cullins are subtle and in no way comparable to the effects of neddylated cullins on HHARI and TRIAD E3 ligase activity. Thus, to address this comment, we added the following sentence to the revised manuscript: “Cells expressing versions of TRIAD1 and HHARI mutated in their RING2 domains exhibited increased steady state levels of neddylated cullin and protected those CRLs from de-neddylation in the presence of MLN4924, although future studies will be required to determine the mechanisms by which TRIAD1 and HHARI impact de-neddylation.”

3. *Although the biochemical work showing the increased activity of Triad1 and HHRAI upon binding is convincing, the biological significance is still unclear. Most of the striking effects are only seen by catalytically inactive mutants. For example, the deneddylation assay in Fig 7C, and the neddylation assay in Supplemental Fig 5D, show no effect of wildtype proteins. Even the effect on Cull1 substrates are only striking for the C357S mutant of HHRAI. Reading the text, it is not clear how the activation of one E3 Ligase (HHRAI) by a second E3 (Cull1) leads to the accumulation of the Cull1's substrates. There appears to be no evidence for increased ubiquitylation or degradation of Cull1 or its substrates as by HHRAI. Also, HHRAI and Triad1 do not appear to be substrates of Cull1 or Cul5. It is not intuitive how activation of Ariadne proteins by cullins leads to regulation of cullin substrates based on the presented data. In the absence of Ariadne substrates, it is impossible to know the effect of cullins beyond that fact that they clearly bind and activate Ariadne subfamily proteins in vitro.*

Response: At the current stage we can only speculate about the mechanisms by which HHARI or TRIAD1 regulate cullin neddylation and activity and we discussed different mechanisms in our manuscript. As pointed out by the referee, future studies will heavily depend on the identification of Ariadne substrates, which will provide the starting point for studying the effects on cullins in more detail.

Detailed comment:

Supplemental Fig 5E. The cell synchronization data are not very convincing to the eye for G1 arrest. The authors should include percentages on the Flow graphs.

Response: HEK293 cells are inherently difficult to synchronize. The referee correctly spotted that cells are not significantly arrested in G1 after serum starvation, which was confirmed by determining the percentages of cell cycle phases (which have been added as requested). Hence, we decided to remove the data regarding serum starvation, but kept the data for thymidine block and nocodazole arrest. Importantly, this does not affect our overall conclusions, which are still valid, suggesting that cells expressing mutant HHARI(C357S) are defective in CUL1-mediated substrate degradation, such as cyclin E and p27^{Kip1}, during cell cycle progression.